# Review of Chitosan-Based Polymers as Proton Exchange Membranes and Roles of Chitosan-Supported Ionic Liquids

**DOI:** 10.3390/ijms21020632

**Published:** 2020-01-17

**Authors:** Nur Adiera Hanna Rosli, Kee Shyuan Loh, Wai Yin Wong, Rozan Mohamad Yunus, Tian Khoon Lee, Azizan Ahmad, Seng Tong Chong

**Affiliations:** 1Fuel Cell Institute, Universiti Kebangsaan Malaysia, UKM Bangi 43600, Selangor, Malaysia; adierahanna@gmail.com (N.A.H.R.); waiyin.wong@ukm.edu.my (W.Y.W.); rozanyunus@ukm.edu.my (R.M.Y.); 2Department of Chemistry–Ångström Laboratory, Uppsala University, Box 538, SE-751 21 Uppsala, Sweden; edison_tiankhoon@hotmail.com; 3Faculty of Science and Technology, Universiti Kebangsaan Malaysia, UKM Bangi 43600, Selangor, Malaysia; azizan@ukm.edu.my; 4College of Energy Economics and Social Sciences, Universiti Tenaga Nasional, Jalan IKRAM-UNITEN, Kajang 43000, Selangor, Malaysia; Stchong@uniten.edu.my

**Keywords:** chitosan biopolymer-based membrane, ionic liquids, performance of fuel cell, fuel cell applications

## Abstract

Perfluorosulphonic acid-based membranes such as Nafion are widely used in fuel cell applications. However, these membranes have several drawbacks, including high expense, non-eco-friendliness, and low proton conductivity under anhydrous conditions. Biopolymer-based membranes, such as chitosan (CS), cellulose, and carrageenan, are popular. They have been introduced and are being studied as alternative materials for enhancing fuel cell performance, because they are environmentally friendly and economical. Modifications that will enhance the proton conductivity of biopolymer-based membranes have been performed. Ionic liquids, which are good electrolytes, are studied for their potential to improve the ionic conductivity and thermal stability of fuel cell applications. This review summarizes the development and evolution of CS biopolymer-based membranes and ionic liquids in fuel cell applications over the past decade. It also focuses on the improved performances of fuel cell applications using biopolymer-based membranes and ionic liquids as promising clean energy.

## 1. Introduction

The occurrence of global warming is due to carbon dioxide emissions, which leads to earth’s climate change and consequently affects the environment, health, and economy. In this context, fuel cells are attracting interest. These cells are devices for power generation that operate in a similar manner as conventional batteries and that require a continuous supply of fuel. They are electrochemical devices that function as good alternative nonpolluting power sources. They generate electricity by using fuel sources such as hydrogen through electrochemical reaction. Fuel cells are among the pollution-free clean energy technologies that meet the requirements for energy conservation, environmental sustainability, and economic growth.

Polymer electrolyte membrane fuel cells (PEMFCs) are highly promising fuel cells for large-scale use; in this technology, an advanced plastic electrolyte in the form of a thin, ion-permeable sheet is used to exchange protons from the anode to the cathode [1]. PEMFCs are light, highly efficient, and compact. They operate at relatively low temperatures (≈80 °C), have high power density, and are suitable for various applications. Figure 1 shows that PEMFC consists of various important elements, namely, bipolar plates, diffusion layers, electrodes (anode and cathode), and an electrolyte. The core of a PEMFC is a membrane electrode assembly (MEA) composed of a proton exchange membrane (PEM) placed between two electrodes.

Nafion is popularly used as a proton-conducting polymer membrane in fuel cells because of its ability to conduct protons, high thermal and mechanical stability, and selectivity for cations [2]. However, Nafion membranes have limitations, such as high manufacturing costs, not eco-friendly, and hydrogen crossover from anode to cathode that reduces fuel efficiency and fuel cell open circuit voltage (OCV) [3]. Nafion also limits the operation of PEMFC to low temperatures (≈80 °C), because it needs hydration and permanent gas humidification to assure high proton conductivity and as the temperature is increased above 100 °C, the water affinity and mechanical strength of the membrane will also be reduced. These limitations of Nafion membranes have prompted researchers to study and produce new environmentally friendly and economical PEM materials from renewable sources, with improved chemical and thermal stability, water uptake, and mechanical properties, that can potentially replace the synthetic polymers in fuel cell applications.

Numerous studies have been carried out to replace Nafion membranes because of their shortcomings, and biopolymer-based composites are considered for environmental protection. Biopolymers are molecules that have long chains and are created from repeating chemical blocks, have enticing properties, and research on biopolymer membrane materials has focused on the development and productivity of biopolymers and increasing their membrane-processing capacity and operation [4]. However, biopolymers such as polysaccharides have drawbacks such as their hydrophilicity properties that can lead to excessive water uptake and swelling in the membrane, which can affect its durability and as the biopolymer membranes are incorporated with super acid that is highly soluble in water, they tend to leach out from the membrane, which then will restrict the long-term use in fuel cell operations [5,6]. Other than that, aside from excellent thermal stability, another factor that affected the fuel cell performance is the durability of the membranes in oxidative conditions. As the cell test is conducted in a strong oxidizing environment, the oxidative degradation is a crucial factor in membrane degradation mechanisms due to the decomposition of products of HO* and HOO* radicals initiated from the hydrogen peroxide that has strong oxidizing characteristics throughout the fuel cell operation. The oxidative stability of the membranes was evaluated through immersion of membranes in Fenton’s reagent for 1 h at 80 °C, and membranes that can endure the oxidative environment can be further used during fuel cell operation. Despite having these drawbacks, polysaccharides have been widely used and studied due to their distinctive properties, abundance, hydrophilicity, and can be chemically modified and functionalized to be used as PEM in fuel cell applications.

Polysaccharides, such as chitosan (CS) and cellulose, are among the best natural polymer materials because of their abundance in the environment. CS is a linear polysaccharide consisting of randomly distributed β-(1-4)-linked ᴅ-glucosamine (deacetylated unit) and N-acetyl-ᴅ-glucosamine (acetylated unit), which is produced by the deacetylation of chitin. CS has high water affinity properties as there is the presence of three different polar functional groups, which are hydroxyl (–OH), primary amine (–NH_2_), and C–O–C groups. CS has a structure that is very similar to cellulose, but the difference among them is that CS contains amine groups and its structure is characterized by its molecular weight and degree of deacetylation, where its solubility is depended on [7].

CS has rigid structure and high crystallinity as there are three hydrogen atoms strongly bonded between the amino and hydroxyl groups within the CS monomers, due to the immobilized structure, which leads to its proton conduction limitation. Moreover, CS has attractive physicochemical properties in aqueous solution due to its polycationic nature and this leads to wide range of biological purposes such as antimicrobial activity and disease resistance activities in plants [8]. CS membranes are one of the most preferable and favorable materials to replace Nafion membranes as they have shown improved performance in low temperature fuel cell applications and due to their enticing alcohol barrier properties, proton conductivity, and thermal stability after they undergo the cross-linking process [5,9].

This review paper focuses on the development and recent progress in the modifications of CS-based composite membranes and the utilization of ionic liquids as proton conductors in composite membranes. We discuss the improvements achieved through the addition of inorganic fillers, CS/polymer blends, and chemical modifications to CS composite membranes. Then, we discuss ionic liquids and their roles in enhancing the proton conductivity and hydrophilicity of CS membranes.

## 2. Biopolymer-Based Membranes

### 2.1. CS-Based Biopolymer Membranes

CS is a natural polymer that is hydrophilic, biodegradable, and biocompatible. It is the N-deacetylated derivative of chitin and the second most abundant natural biopolymer. It can be obtained by deproteinization or by CS alkaline deacetylation of crustacean shells, molluscan organs, fungi, and insects [10]. The hydrophilicity of CS is a desirable property for environments with high temperature and low relative humidity (RH). CS has a high potential for development into sophisticated functional polymers, because its backbone contains free amine and hydroxyl groups. CS is a preferred biomaterial membrane for ultrafiltration, reverse osmosis, and pervaporation because of its low-cost production. CS has attracted attention in diverse scientific and engineering processes given its excellent nontoxicity, eco-compatibility, chemical and thermal stability, and natural abundance [11].

Given that the membrane is the crucial component of PEMFC, it needs to exhibit desirable properties, such as high hydrophilicity, good chemical and thermal stability, to achieve high efficiency [12]. Despite its advantages, CS has low mechanical strength and proton conductivity; it is also highly brittle due to its high glass transition temperature [13]. These disadvantages can be overcome by blending CS with other polymers, creating CS based organic–inorganic hybrids, doping inorganic fillers in the CS matrix, or chemically modifying the CS [14]. CS modified through various techniques, such as sulphonation, phosphorylation, quaternization, and chemical cross-linking, is a cost-effective biopolymer electrolyte membrane with low methanol permeability and suitable ion conductivity, especially at high temperatures [15].

### 2.2. CS-Based Composite Membranes

#### 2.2.1. CS/Inorganic Filler Composite Membranes

Nonmodified CS membranes have high hydrophilic properties that cause a high degree of swelling and reduce sturdiness in fuel cell applications by increasing the water uptake excessively and affecting their fragility [10]. Other than that, in contrast to Nafion membranes, which have high mechanical and thermal strengths and high proton conductivity, the pristine, unmodified CS membranes possess low proton conductivity due to the absence of mobile hydrogen ions in their structure and the excess swelling properties of the membranes gives negative impacts in their mechanical properties. These characteristics limit their function and performance, thus, the CS membranes were modified by incorporating the inorganic fillers into them [13].

Tripathi and Shahi [16] reported that one of the ways to overcome these problems is to make chitosan-chitosan-inorganic composite membranes, because they balance the hydrophilic and hydrophobic nature of chitosan and reduce fuel crossover occurrence that diminishes the membranes’ methanol permeability, enhances the mechanical and thermal stabilities, and improves the proton conductivity and fuel cell performance by incorporating solid inorganic proton conductors. Two strategies for combining inorganic and polymer substances have been summarized from previous studies, namely, the in situ formation of inorganic particles within the biopolymer matrix through the sol–gel reaction or crystallization and the physical mixing of organic solutions and inorganic fillers succeeded by simple casting. Figure 2 shows that sol–gel reaction involves two consecutive steps, namely, the hydrolysis of metal alkoxides to produce hydroxyl groups and the polycondensation of hydroxyl groups to form a 3D network. Inorganic fillers used in CS-based membranes include hygroscopic oxides, heteropoly acids, carbon nanotubes (CNTs), and graphene oxide (GO) have been studied extensively, as summarized in the following subsections.

##### (1) Hygroscopic Oxides

Embedding nonporous and porous inorganic hygroscopic fillers, such as silica, titanium dioxide, montmorillonite, zeolites, metal phosphates, and metal oxides [17], that have permissible pore sizes and structures within membranes, enhances mechanical and thermal strengths and overcomes methanol crossover because these materials could obstruct polymer chain packing and generate diffusion paths with high complexity [10]. The addition of hygroscopic oxides, e.g., Al_2_O_3_, TiO_2_, ZrO_2_, and (SiO_2_)_n_, into the CS polymer matrix by functionalizing the inorganic fillers enhances the water retention capacity and ionic conductivity of polymer composites at low RH as new proton pathway is formed [18,19,20]. Other than that, the methanol crossover can be reduced as the swelling behavior of the membrane is suppressed and the mechanical, chemical, and thermal properties can also be improved through the inherent interfacial interactions between the inorganic fillers and the CS polymer matrix. The study regarding polybenzimidazole membranes with functionalized montmorillonite, which is the protonated montmorillonite (MMT-H) has proven that with very high loadings of MMT-H, the proton conductivity of the membranes achieve drastic improvement at 20% RH with the values of 436 mS cm^−1^ and mechanical strength of 8.4 MPa [21]. Although hydrophilic fillers provide numerous hydrogen bonding sites to increase the water uptake in membranes, conductivity decreases with increasing filler content due to the poor and weak interfacial interaction between the organic polymer and inorganic fillers in hybrid membranes [22,23]. To enhance the morphology, water retention capacity, and proton conductivity of the hybrid membranes and to overcome fuel permeation, inorganic fillers are modified with functional groups, such as sulphonic acid groups and amine or grafted with macromolecules [24,25].

Silicon dioxide (SiO_2_), also known as silica, is a common hygroscopic inorganic filler that is widely used in fuel cells, particularly in direct methanol fuel cells and proton exchange membrane fuel cells. Silica is hydrophilic and is an enticing material for preparing proton exchange membranes because of the support it provides and its thermal stability [26]. A nanostructured N-p-carboxy benzyl CS–silica–PVA hybrid membrane has been prepared through the sol–gel method in aqueous media and cross-linking with HCHO and H_2_SO_4_; afterward, the oxidation of the thiol group into the sulphonic acid group is completed [27]. Proton conductivity increased with increasing content of N-p-carboxy benzyl CS–silica because of high functional group concentration. The –COOH grafted on CS and the –SO_3_H grafted on silica acted as proton-conducting groups. –SO_3_H groups are highly acidic. Thus, membrane conductivity improved when the concentration of –SO_3_H groups was less than the concentration of the acidic –COOH groups. The highest conductivity was obtained when the content of N-p-carboxy benzyl CS-silica was increased to 5.31 × 10^−2^ S cm^−1^ because of the increase in –SO_3_H concentration and the low cross-linking density amongst the fabricated polyelectrolyte membranes. The stability test was performed for 1 h at 80 °C by using Fenton′s reagent (3% H_2_O_2_ aqueous solution containing 3 ppm FeSO_4_) and has shown that in oxidative condition, the weight loss was inhibited due to the highly cross-linked and compact structure, which led to excellent oxidative stability of the membrane [27].

By contrast, the proton conductivity of the composite membrane decreases when unmodified silica particles are incorporated in the biopolymer membrane [21]. Nonetheless, upon functionalization with carboxylic, sulphonic, and quaternary groups on silica, proton conductivity increases due to the increase in water adsorption. The proton conductivity of the carboxylated silica/CS was highest and increased to 0.029 S cm^−1^ by 40% when compared with those of pure unmodified CS, sulphonated silica/CS (0.024 S cm^−1^) and quaternary aminated silica/CS (0.018 S cm^−1^) membranes. Incorporating phosphorylated hollow mesoporous silica sub-microspheres (PHMSS) with amino tris (methylene phosphonic acid) into a CS matrix enhanced water uptake by 47.8%, suppressed swelling properties by approximately 27.1%, and increased the proton conductivity (2.2 × 10^−5^–1.2 × 10^−2^ S cm^−1^ at 80 °C and 41–78% RH) of the hybrid membranes [28]. The mesoporous structure promotes water retention by adsorbing the moisture inside the mesopores [29,30], whilst –PO_3_H_2_ groups successfully improve proton conduction. At 110 °C and 100% RH, the CS/PHMSS-ATMP hybrid membrane incorporated with 7.5 wt.% PHMSS-ATMP sub-microsphere fillers showed the highest proton conductivity of 9.40 × 10^−2^ S cm^−1^, which was higher than those of pristine CS membrane and CS/HMSS hybrid membranes without modification and inorganic sphere incorporation (Figure 3).

In recent years, numerous studies have been conducted on the use of fillers with zwitterionic groups to enhance the proton conductivity of membranes. Novel hybrid membranes of CS/titania–silica inorganic dopant functionalized with carboxyl and amino groups (TiC–SiN) have been fabricated through a facile in situ sol–gel method [31]. The membranes were cross-linked with H_2_SO_4_ solution for 24 h. The inorganic materials from the dopants, titania and silica, which acted as precursors, were functionalized with –COOH and –NH_2_ groups on TiC–SiN dopants and provided zwitterionic groups that could produce new proton pathways to improve the membranes’ proton conductivity and water uptake. The water uptake and proton conductivity of all the hybrid membranes increased with increasing content of TiC–SiN due to the hygroscopic properties of silica and titania and the hydrophilic groups of –COOH and –NH_2_ in TiC–SiN that attracted water. The highest proton conductivity of the CS/2TiC–1SiN−7 membrane (0.0408 S cm^−1^) at 25 °C was four times higher than that of the pristine CS membrane (0.011 S cm^−1^). Proton conductivity also increased with increasing temperature from 60 to 100 °C. This response was due to the zwitterion-functionalized TiC–SiN dopants. These dopants were homogeneously dispersed in the CS matrix and with the aid of acidic groups as proton donors and basic groups as proton receptors generated new proton transfer pathways, which were linked by hydrogen-bonded water molecules through the migration of protons via the Grotthuss mechanism.

Polyaniline is a highly conducting polymer that is used because of its excellent stability, redox reversibility, and unique electrochemical properties. Polyaniline was modified with hydrophilic silica to enhance the morphology and proton conductivity of a hybrid membrane at low water content [26]. Polyaniline/nanosilica (PANI/SiO_2_) was produced as a modified inorganic filler through a facile ultrasonication method. Then, it is incorporated into novel ionic cross-linked CS-based hybrid nanocomposites to produce a CS–PANI/SiO_2_ hybrid membrane that was ionically cross-linked with sulphuric acid [26]. The CS–PANI/SiO_2_ hybrid membranes exhibited improved mechanical properties, oxidative stabilities, and proton conductivity because of the additional proton pathways contributed by the doped polyaniline. The water uptake and proton conductivity of the hybrid membranes increased with increasing content of the nanofiller PANI/SiO_2_ up to 3 wt.% loading but decreased with the further addition of PANI/SiO_2_ fillers because of filler agglomeration, reduced inflexibility, and poor filler dispersion. The proton conductivity of the hybrid membrane decreased when the PANI/SiO_2_ filler loading content exceeded 3 wt.% because of the block effect that inhibited proton transfer or high tortuosity through the proton-conducting pathway [32,33]. The CS–PANI/SiO_2_-3 hybrid membrane showed increased water uptake (57.62–72.87%) and possessed the highest conductivity of 8.39 × 10^−3^ S cm^−1^ at 80 °C; in its hydrated state, its conductivity decreased when the temperature exceeded 80 °C (Figure 4). The highest conductivity shown by the CS-PANI/SiO_2_-3 membrane may be due to the existence of sulphate ions in the doped polyaniline, hydroxyl groups on the surface of silica, and protonated amine groups in the matrix polymer of the membrane; moreover, the higher proton conductivity of CS-PANI/SiO_2_ membranes than that of CS membrane could be due to the incorporated PANI/SiO_2_ fillers, which acted as connecting bridges that shortened the distance of proton hopping [32,33]. Other than that, the stability test was performed by using Fenton’s reagent (3% H_2_O_2_ with 2 ppm FeSO_4_) for 1 h at 80 °C and has shown that as the concentration of PANI/SiO_2_ nanofillers was increased, the weight loss was reduced, and this implied that CS-PANI/SiO_2_ composite membrane has excellent oxidative stability. Consequently, this membrane was tested in single cell for DMFC application and exhibited the peak density of 56.27 mW cm^−2^, higher than the unmodified CS membrane (41.15 mW cm^−2^).

Zeolite is another inorganic filler that is commonly used in polymeric membranes; it represents aluminosilicates with framework structures enclosing cavities that could be occupied by water molecules or large ions [19]. The regular pore size, ordered structure, hydrophilic/hydrophobic nature, excellent stability, and tolerable proton conductivity of zeolite have prominent effects on the properties of CS/zeolite membranes [34,35]. The appropriate addition of zeolites enhanced mechanical and thermal strengths because of the presence of hydrogen bonds between CS and zeolites, whereas the excessive addition of zeolites reduced the mechanical strength of the hybrid membranes because of the formation of excessive interfacial voids at the interface of CS and zeolites [35,36]. The diffusion resistance of methanol increased due to the association of hydrophobic zeolites and consequently decreased methanol permeability, whereas hydrophilic zeolites exhibited the opposite trend [36,37]. The homogenous incorporation of sorbitol as a plasticizer and the proper control of membrane formation temperature could improve the interfacial morphology of CS/zeolite membranes [35]. Functionalizing zeolite particles contributed to the inhibition of a methanol crossover [17,34,37]. The methanol permeability of sorbitol-plasticized CS/mordenite hybrid membranes in 12 mol L^−1^ methanol water solution at 25 °C was 4.9 × 10^−7^ cm^2^ s^−1^, which was 44% lower than that of pristine CS control membrane and four times lower than that of Nafion 117 membrane (2.3 × 10^−6^ cm^2^ s^−1^) under the same test conditions [35].

Surface-modified Y zeolite-filled CS hybrid membranes were fabricated [34] with inorganic fillers modified using the silane coupling agents 3-aminopropyl-triethoxysilane (APTES) and 3-mercaptopropyl-trimethoxysilane (MPTMS). Then, the mercapto group on MPTMS-modified Y zeolite was further oxidized into the sulphonic group (–SO_3_H). The conductivity of the modified Y zeolite-filled CS hybrid membranes increased to 2.58 × 10^−2^ S cm^−1^ with the functionalization of -SO_3_H groups onto the zeolite surface. Moreover, even though the zeolite content increased and the proton conductivity of the hybrid membranes decreased, the proton conductivity of the membranes remained more acceptable (>1.5 × 10^−2^ S cm^−1^) than fully hydrated Nafion 117 (6.91 × 10^−2^ S cm^−1^). The incorporation of inorganic fillers and Y zeolites into the CS matrix further reduced methanol permeability because of the dispersion of Y zeolites, which increased the methanol permeation path length and irregularity and the local rigidification of the CS matrix due to the incorporation of rigid zeolite; thus, membrane swelling and methanol uptake decreased [34].

CS/zeolite beta hybrid membranes were prepared and fabricated, and their performances were enhanced by incorporating zeolite beta particles into the CS matrix. As shown in Figure 5, zeolite beta particles were sulphonated through three different pathways. The presence of –SO_3_H groups on the zeolite surface improved the interaction between the CS and zeolite beta matrix, reduced methanol permeability, and improved the ionic conductivity of CS/zeolite hybrid membranes [17,37]. Zeolite beta was selected as an inorganic filler, because its hydrophobicity could ensure low methanol crossover [36], and its moderate surface acidity could enhance bonding with the CS matrix [38].

CS/Beta hybrid membranes have higher thermal stability than pure CS membranes [11] because of the hydrogen bonding interaction between the –OH groups on the surfaces of zeolite beta and the –OH or –NH_2_ groups of CS [36]. The thermal stability of CS/Beta hybrid membranes increased moderately with increasing zeolite beta contents, because hydrogen bond interactions were enhanced. Moreover, methanol permeability further decreased with the incorporation of zeolite beta particles into the CS matrix. The methanol permeabilities of CS/Beta-2-30% at methanol concentrations of 2 and 12 mol L^−1^ were 7.04 × 10^−6^ and 2.46 × 10^−6^ cm^2^ s^−1^, respectively, which were the lowest amongst all fabricated hybrid membranes because of the extension of diffusion path length of methanol in the membrane, the rigidification of CS chains, and the compression of volume amongst CS chains upon the introduction of zeolite beta particles. Thus, the methanol crossover of CS/Beta hybrid membranes decreased. When zeolite beta particles were incorporated into the CS matrix, water and methanol uptake decreased, and proton conductivity decreased to a certain level. CS membranes filled with γ-glycidoxypropyltrimethoxysilane (GPTMS)-modified zeolite beta particles, in which the uniform zeolite beta was synthesized through a hydrothermal method and then functionalized with GPTMS, have been studied and fabricated [31]. These CS/Beta-GPTMS hybrid membranes exhibited low methanol permeability. The lowest methanol permeabilities of 4.4 × 10^−7^ and 2.2 × 10^−7^ cm^2^ s^−1^ were obtained at the methanol concentrations of 2 and 12 mol L^−1^, respectively. The decrease in the methanol permeability of CS/Beta-GPTMS hybrid membranes was due to the same reasons for such a decrease in CS/Beta hybrid membranes [17]. The water and methanol uptake of CS/Beta and CS/Beta-GPTMS hybrid membranes further decreased with increasing content of zeolite beta incorporated in the hybrid membranes. The possible reasons for this phenomenon are the higher hydrophobicity of zeolite beta compared with CS and the rigidification of CS chains upon the addition of zeolite beta, thereby reducing their ability to adsorb solvent molecules. Additionally, the proton conductivity of CS/Beta-GPTMS hybrid membranes decreased further as the zeolite beta content of the hybrid membranes was increased. The highest proton conductivity of the CS/Beta-GPTMS-3% hybrid membrane was 1.46 × 10^−2^ S cm^−1^, which was slightly lower when than that of the CS/Beta-3% hybrid membrane (1.53 × 10^−2^ S cm^−1^). The existence of zeolite beta particles in the hybrid membranes extended the proton transport path. Thus, proton conductivity of the CS/zeolite beta hybrid membranes was lower than that of the pristine CS membrane. In addition, the proton conductivities of CS/Beta-GPTMS hybrid membranes were lower than those of CS/Beta hybrid membranes because of their low water uptake [37].

Previous works by various researchers have proven that the CS composite membrane with unmodified inorganic hygroscopic oxide fillers possess low proton conductivity, water retention capacity, and mechanical stability of the membrane. Thus, these drawbacks can be solved through the presence of modified hygroscopic oxide fillers, such as functionalized silica and zeolite, in CS hybrid membranes, due to hydrogen bonds existing between the CS polymer and the fillers, as well as the presence of the zwitterionic groups in the fillers, which provide an additional proton-conducting pathway in the membrane. These occurrences have suppressed fuel crossover and improved the morphology, proton conductivity, water retention capacity, and mechanical strength and consequently enhanced fuel cell performance of the composite membranes. Table 1 shows a summary of applications of hygroscopic oxide fillers in CS hybrid membranes for fuel cells.

##### (2) Heteropoly Acids

Solid superacids have been extensively studied in the last 20 years. Gillespie described superacids as acids with acidity exceeding that of 100% pure sulphuric acid; most superacids are prepared through combination of strong Lewis and Bronsted acids [44]. Solid acids and superacids have been actively used and prepared for studies on dehydration, cracking, hydrocracking, isomerization, alkylation, and acylation [45]. Solid superacids are categorized into three types, namely, metal oxide-supported sulphates (M_x_O_y_-SO_4_^2−^), heteropoly acids (HPAs), and zeolite solid superacids. These acids may enhance the proton conductivity and mechanical properties of CS membranes because of their satisfactory mechanical, hygroscopic, and proton conductive properties [37].

HPAs have been used in fuel cell applications to improve proton conductivity. HPAs are strong Bronsted acids, solid electrolytes, soluble in polar solvents, and solid crystalline materials that are highly conductive and thermally stable; moreover, they exhibit well-defined molecular structures, surface charge densities and chemical and electrochemical properties [46]. HPAs are acids that are made from the combination of hydrogen and oxygen with certain metals and nonmetals; they may adopt the Keggin structure (H_n_XM_12_O_40_) and the Dawson structure (H_n_X_2_M_18_O_62_) because of the possibilities of different combinations of addenda atoms and different types of hetero atoms. HPAs are highly solubility in water, and their leakage during cell operation may degrade fuel-cell performance [47,48,49]. This obstacle can be solved by mixing HPAs, such as phosphotungstic acid (H_3_[P(W_3_O_10_)_4_]), phosphomolybdic acid (H_3_[P(Mo_3_O_10_)_4_]), and silicotungstic acid (H_4_[Si(W_3_O_10_)_4_]) with the CS matrix to form insoluble complexes that could be immobilized within membranes through strong electrostatic interactions [50,51]. HPAs exhibit strong proton conductivities that could reach as high as 1.9 × 10^−1^ S cm^−1^ at room temperature, thereby helping improve the proton conductivity of the CS polymer matrix [52].

The ionic conductivities of CS–phosphotungstic acid (CS/PTA) composite membranes proposed and tested as proton conductors in low-temperature fuel cells fed by methanol, borohydride, and hydrogen are higher than that of CS membranes [53,54]. CS/PTA polyelectrolyte membranes prepared using porous alumina medium for the slow release of H_3_PW_12_O_40_ exhibited good performance when applied as electrolytes in H_2_ feed fuel cells, and a proton conductivity of ≈14 mS cm^−1^ was achieved [55,56,57]. The addition of a drying step on a glass support reduced the shrinkage and fabrication of flat homogeneous membranes and improved performance; the measured peak power density of membranes prepared through this approach was 550 mW cm^−2^ [57]. CS/phosphomolybdic acid (CS/PMA) membranes exhibited a maximum peak power density of 60 mW cm^−2^ with a proton conductivity of 1.6 mS cm^−1^. However, free-standing phosphomolybdic acid and mixed phosphotungstic/phosphomolybdic acid CS membranes fabricated and functionalized for 24 h exhibited a higher peak power density of 350 mW cm^−2^ with a proton conductivity of ≈7 mS cm^−1^ relative to the CS/PMA membranes when both membranes were tested as proton conductors in a low temperature H_2_-O_2_ fuel cell [58].

The mesoporous cesium salt of phosphotungstic acid has been embedded in CS hybrid membranes [59]. The mesoporous properties of the cesium salt of PTA facilitate strong and maximized intermolecular interaction between the organic and inorganic phases in the membranes and enhanced the dispersion degree. At the optimal loading of m-PTA, the maximum peak power density reached 83 mW cm^−2^ with then proton conductivity of 0.0185 S cm^−1^ when the membranes were fed with 2 M methanol fuel. The m-PTA also reduced the surface tension originating from the solid substance and increased the mechanical stability of the membrane. Tensile strength increased from 56.43 to 64.30 N mm^−2^ as the loading of m-PTA was increased from 1 to 10 wt.%. In addition, CS/m-PTA hybrid membranes are attractive alternative membranes for use in direct methanol fuel cells, because they are inexpensive, easy to fabricate, environmentally friendly, and could operate efficiently at high methanol concentrations.

CS and silica-supported silicotungstic acid (IHPA) nanocomposite membranes were produced by using a simple solution casting method [60]. The silica-supported silicotungstic acid-incorporated membranes, also known as inorganic HPA, showed improved thermal, mechanical, and oxidative stabilities and selectivity because of the strong electrostatic interaction and hydrogen bond between the polymer, cross-linking agent (0.5 M H_2_SO_4_), and IHPA nanoparticles (Figure 6). The proton conductivities of the CS–IHPA membranes were measured at different temperatures ranging from 30 to 125 °C under dry and hydrated conditions. The proton conductivity of all membranes increased with increasing temperature, because the flexibility of polymer chains and the mobility of water molecules increase at elevated temperatures. The highest proton conductivities of the CS-IHPA-5 membrane in the hydrated state at 30 and 90 °C were 5.8 × 10^−3^ and 9.0 × 10^−3^ S cm^−1^, respectively, which were higher than those of the pristine CS membrane (1.07 × 10^−4^ S cm^−1^). The increase in the proton conductivity of the membranes was ascribed to the presence of –OH groups on silica, which yielded additional conduction sites for proton transfer; moreover, the interaction between silicotungstic acid and silica in IHPA resulted in the formation of a continuous hydrophilic channel that permitted the transfer of protons in the form of hydronium ions through diffusion [61,62,63].

The CS-IHPA-5 membrane exhibited comparatively higher water bonds than other membranes, and its proton conductivity increased gradually with increasing temperature because of its satisfactory water retention capability [64]. Nevertheless, when the filler concentration exceeded 5 wt.%, and the temperature exceeded 90 °C, proton conductivity noticeably decreased. Excess filler loading (>5 wt.%) resulted in agglomeration in the CS-IPHA-7 membrane, thereby reducing the effectiveness for proton conduction by IHPA. The block effect caused by the reduced space between the IHPA filler particles when the IHPA content was excessively high (7 wt.%) inhibited proton transfer or generated proton-conducting pathways with high tortuosity, hence decreasing proton conductivity. The stability test was performed by immersing the membrane in Fenton’s reagent for 1 h at 80 °C, which has shown the composite membrane has better oxidative stability than pure unmodified chitosan, which more than 98% weight of the composite membranes were sustained after conducting the test. This occurrence was due to the shielding effect from the polymer chain and cross-linking agent presented in the composite membrane that prevented the free radicals of hydroxyl and hydro peroxyl from attacking the polymer chains. Moreover, the CS-IHPA-3 membrane exhibited a higher OCV of 0.73 V due to the addition of IHPA and a maximum power density of 54.2 mW cm^−2^ at a current density of 153.7 mA cm^−2^ when compared with the pristine CS, which has a maximum power density of 41.2 mW cm^−2^ and a current density of 134.4 mA cm^−2^ in a single-cell DMFC under fully hydrated conditions at 80 °C (Figure 7) [60].

The high solubility of HPAs in electrochemically produced water promotes leakage and decreases performance, but mixing the HPAs into a CS matrix can alleviate these issues. Besides that, the fuel cell performance and proton conductivity, as well as the mechanical properties of the CS-HPAs composite membranes can be further improved as the HPAs are functionalized by adding inorganic fillers and mesoporous nature onto them, which have the roles of providing proton conduction sites and the degree of dispersion between the polymer matrix and the modified HPAs that led to the formation of homogeneous hybrid polymer membranes. Table 2 shows the summary of HPA-related CS hybrid membranes in fuel-cell applications.

##### (3) Carbon Nanotubes

CNTs, carbon allotropes with cylindrical nanostructures comprising rolled-up graphene sheets that appear similar to an elongated version of C_60_ were discovered in 1991 [71]. CNTs are composed of carbon atoms linked in hexagonal shapes, with each carbon atom covalently bonded to three other carbon atoms; these materials can be classified as single-walled CNTs (SWCNTs), double-walled carbon nanotubes (DWCNTs), or multiwalled CNTs (MWCNTs) with open or closed ends [71,72]. CNTs can be fabricated through three methods, namely, chemical vapor deposition (CVD) technique [73], laser ablation [74,75], and carbon arc-discharge [76,77,78].

In addition to silica and clay nanoparticles, CNTs have been used as nanofillers for the fabrication of high-performance polymer nanocomposites [79,80], because they have high flexibility, low mass density and extremely high aspect ratio, tensile modulus, and mechanical strength [81,82]. However, polymer/CNT composites are rarely favored, because they exhibit poor dispersion, inert ionic conduction [83], strong π–π interactions, and the ability to form electronic channels in PEMs due to their exquisite electrical conductivity, thereby promoting the risk of short-circuiting in fuel cells. In addition, CNTs must be dispersed uniformly in a polymer matrix considering the strength of the basic Van der Waals interaction amongst tubes. Hence, to overcome these hindrances, the surface of CNTs are functionalized with simple acid groups [84,85] and inorganic materials [33,86], such as carboxylic acid-functionalized CNTs [87], sulphonated CNTs [88], phosphonated CNTs [85] and Naf-, ion-, and polybenzimidazole-functionalized CNTs [89], which are used as inorganic fillers to alter PEMs. Moreover, polyelectrolyte-functionalized CNTs present new proton conducting pathways for rapid proton transport, increased compatibility with polymer matrixes, and additional modification sites [83].

CNTs have been used in composite membranes, and silica-coated CNTs (SCNTs) prepared through a sol–gel method have been applied to the fabrication and preparation of CS/SCNT composite membranes with various SCNT contents [79]. With increasing content of SCNTs from 1 to 10 wt.%, the water uptake of the membranes decreased from 136% to 100% because of the interaction of hydrogen bonds that formed between CSCS and SCNTs [90]. In addition, coating a silica layer containing –SiOH groups on CNTs generated electrostatic and hydrogen bonding interactions between SCNTs and the CS matrix and reduced the moisture absorption capacity of the CS matrix. The proton conductivity of CS/SCNT membranes increased with increasing SCNT content from 0 to 5 wt.% and reached values of 0.015–0.025 S cm^−1^ but reduced to 0.020 S cm^−1^ as the SCNT content was further increased to 10 wt.%. Proton conductivity improved when SCNT contents were less than 5 wt.% because of the fine dispersion of SCNTs, which helped improve the formation of continuous proton transport channels and increased proton concentration [86]. Proton conductivity in a PEM is affected by and depends on effective proton mobility and proton concentration [91]. When SCNT contents are excessive (10 wt.%), the agglomeration of SCNTs in the CS matrix increases the tortuosity of the proton-conducting pathway, thereby decreasing proton conductivity. Besides that, a stability test was performed by immersing the membranes in Fenton′s reagent for 1 h at 80°C, which has shown the CS/SCNTs membrane with higher content of SCNTs has better oxidative stability, which was increased by 86% when compared to the pristine CS membrane. This improvement was due to the excellent resistance and restraint effect of the SCNTs on the CS polymer chains.

CNTs functionalized with CS polyelectrolytes through a facile noncovalent surface-deposition and cross-linking method have been studied [76]. CS-coated CNTs (CS@CNTs) were introduced into the CS matrix, and CS/CS@CNT composite membranes were fabricated. The CS coating could improve the compatibility between CNTs and the matrix, enable the homogeneous dispersion of CS@CNTs, enhance the mechanical properties of the composites, and provide supplementary proton-conducting pathways through the membranes. The water uptake of CS/CS@CNT membranes increased from 93.88% to 108.28% with increasing contents of CS@CNTs from 0% to 1% and then decreased to 93.05% when the CS@CNTs content was increased to 3%. As previously reported [79], water uptake decreased, because the incorporation of excessive CS@CNTs reduced ion channel sizes and moisture absorption capacity [92]. The CS/CS@CNTs membrane with 1% CS@CNTs filler had the highest proton conductivity of 3.46 × 10^−2^ S cm^−1^ at 80 °C amongst the produced membranes. Other than that, the stability test was performed to observe the degradation of the membranes in Fenton’s reagent at 80 °C, which has shown that the CS/CS@CNTs composite membrane with higher content of CS@CNTs has better oxidative stability, enhanced by 62%, indicated that the filler has excellent oxidation resistance and it was compatible with the polymer matrix, as the degradation of the CS polymer chains was inhibited. Single test results and DMFC performances have shown that the OCV and maximum power density of the CS/CS@CNTs-1 membrane increased to 0.72 V and 47.5 mW cm^−2^, whereas pure CS membrane had values of 0.63 V and 36.1 mW cm^−2^, respectively. Considering their simple preparation method and good performance, the fabricated CS/CS@CNT composite membranes are promising PEMs for fuel cell applications [83].

CNTs were coated with superacidic sulphated zirconia (SZr@CNT) through a facile surface-deposition method and incorporated it into the CS matrix [93]. The inorganic proton conductor, superacidic sulphated zirconia (SZr), which has distinguished hydrophilicity, proton conductivity, and stability, was functionalized to CNT to alter PEMs for fuel cell applications. This material successfully improved mechanical strength by 50% and enhanced the proton conductivity (3.4 × 10^−2^ S cm^−1^) of the CS/SZr@CNT membrane compared with those of a pristine CS membrane. When the SZr@CNT filler was added to the CS matrix, the Young′s modulus and tensile strength of the CS/SZr@CNT drastically increased and were, respectively, 118% and 50% higher than those of pristine CS membranes. These improvements resulted in the satisfactory mechanical stability of CNT and the fine and homogeneous dispersion of SZr@CNT in the CS matrix. These results could reduce the mobility of the polymer chains under stress and promote mechanical stability. The proton conductivity of all membranes continuously increased with increasing temperature because of the improvement in water uptake at high temperatures. The highest proton conductivity at 80 °C of the CS/SZr@CNT membrane with 0.5 wt.% SZr@CNT content was 3.40 × 10^−2^ S cm^−1^, which was greater than that of the pristine, unmodified CS membrane (≈10^−4^ S cm^−1^). Moreover, the CS/SZr@CNT membrane with 0.5 wt.% SZr@CNT had a peak power density of 64.6 mW cm^−2^ at 70 °C when tested in single cell and presented great durability after continuous operation for 100 h at the same temperature. The considerable improvement in the performance of the novel CS/SZr@CNT composite membranes verified that these membranes have potential for use as PEMs for fuel cell applications [93].

The poor dispersion of CNTs in the polymer matrix was caused by strong Van der Waals forces, which can be modified by altering the surfaces of the CNTs. CNTs that are functionalized with inorganic materials have proven their ability in providing additional proton-conducting pathways, which helped in enhancing the proton conductivity of the membrane. Other than that, this modification considerably improved the compatibility of CNT fillers with the polymer matrix, producing homogeneous dispersion of CNT-polymer mixture, and thus enhanced the mechanical stability and fuel cell performance of the composite membranes. Table 3 shows the summary of applications of CNTs used in CS hybrid membranes in fuel cells.

##### (4) Graphene Oxide

GO is a single-atomic-layered material that consists of oxygen-containing groups, including carbonyl, hydroxyl, carboxyl, and epoxy groups. GO is an enticing active nanofiller for PEM due to its particular properties of facile modification given the presence hydrophilic oxygen-containing groups [101,102,103,104], good mechanical stability and water retention capacity, and potential to improve the proton conductivity of membranes [105,106,107,108]. Moreover, GO exhibits a large surface area, and its proton conduction could be improved through the hopping mechanism, because it contains hydrophilic functional groups. CS membranes were modified by incorporating GO to produce CS hybrid membranes as a promising approach to enhance the interfacial and mechanical properties, proton conductivity, and fuel cell performance of these membranes.

CS and GO have been prepared through a simple self-assembly method given that both materials are in aqueous condition and used to produce CS/GO nanocomposite films [102]. GO cross-linked CS nanocomposite membranes was first reported and prepared by Shao et al. [103]. The incorporation of GO into the CS matrix improved the interfacial and mechanical properties of CS/GO nanocomposite membranes. SEM images showed that GO was well-dispersed in the CS matrix because of the absence of agglomeration, thereby affirming satisfactory adhesion between the GO filler and CS matrix. In addition, these mechanical properties of CS/GO nanocomposite films had improved relative to those of CS films when various contents of GO were added into the CS matrix. The tensile strength and Young’s modulus of the CS/GO nanocomposite film incorporated with 0–1 wt.% GO loadings increased by 122% and 64% to 40.1–89.2 MPa and 1.32–2.17 GPa, respectively [109]. Meanwhile, at elevated temperature, the cross-linking degree of CS/GO strongly increased with increasing GO loading, and the gel content reached almost 100% upon the addition of 2 wt.% GO, thereby indicating that GO can efficiently cross-link with CS [110]. The tensile strength of the GO cross-linked CS membrane was also enhanced by 141% to 104.2 MPa, which was higher than that of pristine CS membrane (43.2 MPa). GO plays an important role in enhancing the mechanical properties of CS-based nanocomposite films, and the uniform dispersion and cross-linking of CS/GO improves interfacial adhesion and electrostatic attraction of nanocomposites [109,110].

The performance of GO nanofillers could be enhanced through the incorporation of acid groups, such as –SO_3_H. These nanofillers are suitable for mixing with the CS matrix and exhibit homogeneous dispersion, because acid groups could increase the hopping sites for proton migration, thereby increasing the proton conductivity of CS membranes. Similar works have prepared sulphonated GO (SGO) nanosheets with controllable sulphonic acid group loadings [52,111,112] for incorporation into the CS matrix in the production of CS/SGO membranes. The mechanical properties of the CS/SGO membrane improved with increasing SGO filler content from 0 to 4 wt.%, and the tensile strength and Young’s modulus of 44.7–85.3 and 692.3–2697.4 MPa, respectively, were obtained. Moreover, the proton conductivity of CS/SGO membranes was enhanced with increasing content of SGO filler and loading amount of sulphonic acid groups. CS/SGO with 4 wt.% sulphonic acid group and 2 wt.% SGO (CS/S4GO-2) exhibited the highest proton conductivity of 0.0267 S cm^−1^, which was 222.5% higher than that of pure CS membrane (0.0117 S cm^−1^) under hydrated conditions at 25 °C. However, as the SGO filler content was further increased to 2.5 wt.%, proton conductivity decreased because of the agglomeration of nanosheets. The CS/S4GO-2 membrane that displayed the highest proton conductivity also showed improvement in H_2_/O_2_ cell performance with the highest OCV of 0.99 V, the current density of 459.3 mA cm^−2^ and the power density of 146.7 mW cm^−2^ at 120 °C; the OCV value of this membrane was higher than that of Nafion membranes (0.96 V) under similar conditions [52,113].

Another method to improve CS hybrid membranes is by functionalizing CS with –SO_3_H groups and GO filler with –PO_3_H acid-grafted group. This functionalization method enhanced water retention capacity, mechanical properties and proton conductivity, and prevented membrane degradation, given that the functionalized GO filler was assimilated into the CS matrix. The proton conductivity of N-o-sulphonic acid benzyl CS (NSBC) and *N*,*N*-dimethylenephosphonic acid propylsilane graphene oxide (NMPSGO) composite membrane reached the highest value of 8.87 × 10^−2^ S cm^−1^ when 8 wt.% NMPSGO filler was incorporated into the membrane; this value was higher than that of pristine membrane (4.53 × 10^−2^ S cm^−1^) [62]. Furthermore, the same membrane that exhibited the highest proton conductivity also showed the highest water uptake (69.98% at 100% RH), and the water retention capacity of the membrane increased by 4-fold of the retention capacity of the Nafion 117 membrane. The mechanical properties of NSBC/NMPSGO membrane at 30 °C increased when the loading of NMPSGO was increased from 0 to 8 wt.% (≈1651 to ≈2462 MPa). When filter loading exceeded 8 wt.%, the membrane performance decreased, and the structure of membrane became brittle because of the agglomeration of NMPSGO filler in the membrane matrix.

Phosphorylated GO (PGO) nanosheets have been introduced into a CS matrix to produce CS/PGO nanohybrid membranes [114]. The phosphonic acid (PA) group, which acted as a proton carrier that was attached to the GO filler, could conduct the proton structure diffusion of protonic defects through a mechanism identical to the mechanism of proton transfer under water-free conditions [115,116]. The mechanical properties of CS/PGO membranes were enhanced with increasing PGO filler loading from 0.5% to 2.5%, and tensile strength values increased from 49.4 to 51.5 MPa. This increment was ascribed to the suppression of the chain motion of the CS matrix and interfacial interactions with the PGO filler. With increasing PGO filler loading from 0.5 to 2.5 wt.%, proton conductivity increased from 19.0 to 31.0 mS cm^−1^ at 45 °C and 100% RH. The highest conductivity achieved by the CS/PGO membrane with 2.5 wt.% PGO filler was 63.4 mS cm^−1^ at 95 °C. This trend of conductivity increment was due to the mobility of CS polymer chains and water molecules and enlarged free volume. Moreover, the CS/PGO membrane with 1.5 wt.% PGO filler had an OCV value of 0.99 V, maximum current density of 332.5 mA cm^−2^, and maximum power density of 107.0 mW cm^−2^ at 120 °C when tested in a single cell.

In conclusion, previous studies showed that the usage of GO as a filler in CS polymer matrix formed well-dispersed CS-GO composite membranes with good mechanical properties. However, the mechanical properties of the composite membrane can be further improved due to the homogeneous dispersion mixture by adding functionalized GO into the CS polymer matrix. Other than that, this modification included the introduction of acid groups, which acted as proton carriers, providing hopping sites for proton migration in the membranes, enhancing their proton conductivity and fuel cell performance. Table 4 shows the summary of applications of GO in CS hybrid membranes in fuel cells.

#### 2.2.2. CS/Polymer Blend Membranes

Polymer blending is the physical mixing of two or more polymers with or without any chemical bonding between them to obtain polymeric materials with unique properties [5]. Polymer blending is performed to improve the properties and performance and overcome the shortcomings of CS membranes. A new improved CS/polymer blend membrane can be formed when synthetic and natural polymer membranes are combined with CS membranes. Two major approaches are used to prepare CS/polymer blends, namely, the dissolution of polymers in solvent followed by evaporation (solution blending) and the mixing of polymers under fusion conditions (melt blending) [120,121,122]. Solution blending is preferable in CS/polymer blends and often performed with the addition of cross-linking agents through casting to improve the mechanical properties of the blended membranes [123].

Blending CS with synthetic polymers, such as polysulphone (PSF), poly (vinyl alcohol) (PVA), polyethersulphone (PES), and poly (vinylidene fluoride) (PVDF), has been studied. This method is appealing, because synthetic biodegradable polymers blended with CS matrix could help enhance the water retention capacity and mechanical properties of the blended membranes. The tensile strengths and moduli of these synthetic biodegradable polymers are 16–50 and 400–3000 MPa, respectively [124].

CS and PSF are used to prepare blend membranes despite their distinctive characteristics [125,126,127]. CS could contribute essential conducting properties, and its hydrophilicity could be controlled through cross-linking, whilst PSF could provide structural stability to the membrane [126]. To improve the adhesion of CS and PSF, two methods for modifying the surface of PSF membranes have been proposed. The first method is related to the controlled level of the sulphonation of the chemical structure of PSF [128]. A blended membrane was formed by casting a thin layer of CS and microporous sulphonated PSF (SPFS) as shown in Figure 8. This blend membrane exhibits disadvantages because of the weak interaction and adhesion between the two component polymers. As shown in Figure 9, to overcome this weakness, the surface modification of PSF through sulphonation was conducted to form a SPSF/CS blend membrane, which exhibited higher ionic conductivity than pristine CS membrane [127]. The SPSF/CS blend membrane presented the highest ion exchange capacity, water uptake, and proton conductivity of 2.44 mequiv g^−1^, 66.67%, and 0.046 S cm^−1^, respectively. The proton conductivity of this membrane reached 0.180 S cm^−1^ with increasing temperature up to 120 °C, and this value was higher than that of Nafion 117 membrane (≈0.140 S cm^−1^). Thus, the improvement shown by the blend membrane indicated that the sulphonated porous substrate modified into CS matrix efficiently played a supporting role. It promoted proton conduction in the membrane.

CS and PSF are suitable for blending despite their distinctive features, because modified CS could promote hydrophilicity, whilst PSF could improve the structural stability of the blended membranes [125,129]. CS was modified to produce N-phthloyl CS through reaction with phthalic anhydride in di-methylformamide, and the modified CS was then blended with sulphonated polysulphone (SPSF) to form sulphonated polysulphone/N-phthaloyl CS (SPSF/CS) blended membranes [130]. The ion exchange capacity and water swelling of the blended membranes increased to 0.177 mmol g^−1^ and ≈280%, respectively, with increasing concentration of the SPSF. The presence of hydrophilic sulphonic acid groups increased, and ion exchange capacity and water swelling were enhanced with increasing concentration of the SPSF.

In addition, PVA polymers are widely used in polymer blended membranes with CS and have been applied in fuel cell applications [131,132,133,134]. PVA/CS (PCS) blended membranes were prepared then cross-linked with glutaraldehyde and applied to a direct methanol alkaline fuel cell (DMAFC) [131]. This blended membrane exhibited enhanced tensile strength and proton conductivity of 46 N mm^−2^ and 20.0 × 10^−3^ S cm^−1^, respectively, and a low methanol permeability of 6.158 × 10^−7^ cm^2^ s^−1^. Upon blending CS with PVA, mechanical strength, proton conductivity, and methanol permeability improved. A CS/PVA blended membrane was initially modified through quaternization with binary quaternizing agents (hexadecyltrimethylammonium bromide (HDT) and 2,3,5-triphenyltetrazolium chloride (TPTZ)) then mixed together to produce a QPVA/QCS blended membrane [134]. The ion exchange capacity and proton conductivity of the CS/PVA membrane quaternized with HDT were 0.60 mmol g^−1^ and 7.20 × 10^−3^ S cm^−1^, respectively, and these values were considerably higher than those of the membrane quaternized with TPTZ. The single-cell test was operated in an alkaline polymer electrolyte fuel cell using the CS/PVA membrane with HDT cross-linker and provided a peak power density of 15 mW cm^−2^ at 40 °C.

PES is an attractive polymer for blending with CS matrix due to its good mechanical stability, flexibility and low cost [135,136]. However, it has hydrophobic properties, thereby making it insoluble in dipolar solvents. PES has been modified into aryl-substituted sulphonated PES (SPES) to solve this problem. SPES is also blended with N-phthaloyl CS (NPHCs), thereby helping to enhance the hydrophilicity, porosity and water retention of the membrane [129,137]. Muthumeenal et al. [138] reported the increase in proton conductivity and water uptake of SPES/NPHC blended membrane (9.20 × 10^−3^ to 12.10 × 10^−3^ S cm^−1^ and 38.0% to 41.5%, respectively) with increasing temperature from 25 to 80 °C. These increments in proton conductivity and water uptake were due to the presence of sulphonic acid groups in the membrane and polar groups in NPHCs. These findings were supported by Li et al. [137], who also observed increased membrane hydrophilicity with increasing NPHC content.

Vijayalekshmi and Khastgir [139] developed CS/partially sulphonated poly (vinylidene fluoride) (CS/SPVDF) blend membranes and applied them to DMFC. PVDF polymers have been previously used to reduce methanol crossover to a certain limit; however, despite this convenience, PVDF polymers have hydrophobic properties that diminish the ion exchange capacity and proton conductivity of membrane [140,141,142]. This problem can be overcome by modifying PVDF polymers through sulphonation and blending SPVDF with CS, producing a blended membrane with good mechanical strength, water uptake, and retention capacity [143,144]. With increasing content of SPVDF in the blend membrane, water uptake and ion exchange capacity values increased by ≈50% and 0.22 mmol g^−1^, respectively. Furthermore, the mechanical stability improved with increasing SPVDF content, thereby improving the tensile strength of the CS/SPVDF blend membrane under both dry and hydrated conditions (51.8 and 27.6 MPa). The proton conductivity achieved by the blended membrane also increased to 8.12 × 10^−3^–2.85 × 10^−2^ S cm^−1^ with increasing temperature from 30 to 90 °C, respectively. The lowest methanol permeability obtained was 1.8 × 10^−7^ cm^2^ s^−1^, and the CS/SPVDF blend membrane was highly selective (1.69 × 10^4^ S cm^−3^ s). These results proved that this blended membrane was suitable and has potential for use as PEM in fuel cell applications.

It can be concluded that as CS polymers are blended with other synthetic polymers, particular properties of the produced blended membranes, such as mechanical stability, proton conductivity, and water uptake can be improved. However, there are certain polymers that possessed hydrophobic properties, which caused reduction in proton conductivity and ion exchange capacity of the blended membranes. These drawbacks can be promoted by functionalizing the polymers with sulfonic acid groups and as the blended membranes undergo a cross-linking process, the mechanical stability, proton conductivity, water retention capacity, and ion exchange capacity of the blended membranes are further improved.

### 2.3. Chemical Modifications of CS Biopolymer-Based Membrane

Ma and Sahai [10] reported the presence of free hydroxyl and amine functional groups on the backbone of CS, which allowed various chemical modifications of CS, such as sulphonation, phosphorylation, quaternization and chemical cross-linking, to tailor its properties for specific applications. These modifications can make CS a cost-effective biopolymer electrolyte membrane with low methanol permeability and suitable ion conductivity, especially at high temperature [15]. These modifications can also help improve its chemical and mechanical properties and possibly generate ion exchange sites and enhance the ionic conductivity of the membrane [10].

#### 2.3.1. Sulphonation

The modifications of CS through sulphonation have been studied over the years. Sulphonated CS (SCS) was produced through the incorporation of sulphonate groups into CS matrix. Sulphonate groups could be attached to CS matrix in three routes to produce different types of modified CS. The routes included the following: (i) initiation through compounds containing sulphonate groups (R–SO_3_^−^) that produce sulphonated products (–NH–R–SO_3_^−^); (ii) introduction on hydroxyl groups which produce sulphate products (–O–SO_3_^−^); (iii) prompt binding on free amino groups to produce sulphamate products (–NH–SO_3_^−^) [145,146,147,148,149]. Given that SCS derivatives have polyampholytic behavior and specific physicochemical properties, they have been widely used in biomedical fields, such as antimicrobial, antioxidant, antiviral, drug delivery, hemocompatible biomaterial, and bone tissue engineering [150].

Jayakumar et al. [151] reported that a certain density of sulphonate groups can be chemically attached to CS backbone by using various methods and by altering reaction time, temperature, and reactant concentration. N-sulphonated CS (where the sulphonate group is attached to the –NH_2_ sites) and O-sulphonated CS (where the sulphonate group is attached to the –OH sites) were prepared by using sulphating reagents under various reaction conditions. N-sulphonated CS with various sulphonation degrees can be prepared using propane sultone (Figure 10) [152]. Sulphonated CS has a pendant alkyl sulphonic group attached to the side chains. Proper pretreatment conditions and solvent systems enable sulphonation to occur selectively on the C3/C6-position (–OH sites). SO_3_ or chlorosulphonic acid with dimethylformamide complex has been used as a sulphonating reagent [111].

Pure unmodified CS-based biopolymer membranes show low proton conductivity, which can be enhanced via the functionalization of CS with different groups, especially with sulphonic acid, –SO_3_H groups. Xiang et al. [153] prepared sulphonated CS polymer by grafting CS monomers with sulphonic groups and then preparing CS sulphate blending membrane (CCSM). In order to be used in PEMFC, CCSM was blended with pure chitosan at different weight ratios, and cross-linking occurred when the bonding reaction happened between the sulfonic groups in CCSM and the amide groups in the pure chitosan monomers due to the excessive swelling of CCSM. The mechanical properties of CCSM have improved as the tensile strength value increased from 52.9 to 87.8 MPa, indicating that the CS backbone was stable. The proton conductivity and methanol permeability reduction of the advanced CCSM were improved compared with those of pure nonmodified CS membrane, and the highest values of 3.1× 10^−2^ S cm^−1^ at 80 °C and 3.8 × 10^−7^ cm^2^ s^−1^, respectively, were obtained. These improvements exhibited by CCSM showed that CS sulphate successfully functioned as a proton conductor and methanol barrier [153].

Shirdast et al. [111] fabricated a CS/sulphonated CS (CS/SCS) blend membrane by incorporating SCS into the CS matrix. The SCS was prepared using a similar method that was used by Xiang et al. [153], which involves grafting the CS monomers with sulphonic groups. SCS addition to CS matrix caused several enhancements in the performance of the blend membrane. The tensile strength of CS/SCS membrane was improved by 35%, increased from 72.4 to 96.9 MPa. Moreover, the IEC value increased (0.97 mequiv g^−1^) when compared with that of pristine CS membrane (0.65 mequiv g^−1^) because of the presence of the sulphonic acid groups in the sulphonated CS matrix. The CS membrane showed a proton conductivity value of 1.30 mS cm^−1^. However, when blended with 10 wt.% SCS, the CS membrane showed increased proton conductivity that reached 2.20 mS cm^−1^ due to the presence of proton hopping sites of the CS/SCS membrane. The methanol permeability of CS blended with SCS was reduced to 5.91 × 10^−8^ cm^2^ s^−1^, which ensured the efficient application of polymeric membrane [111].

#### 2.3.2. Phosphorylation

Phosphorylated chitin and CS derivatives were applied in several fields, such as biomedical, orthopedics, cosmetics, agricultural, food industrial, and water treatment [154]. Chitin and CS solubility was modified by incorporating phosphonic acid or phosphonate groups through reactions occurring between the phosphorylating agent and the amino groups. Nishi et al. [155] claimed that the solubility of phosphorylated CS can be enhanced even at a low degree of substitution due to the strong interaction between phosphoryl groups and water. This enhancement in solubility indicated that the phosphorylated CS had high hydrophilicity properties and can thus reduce the crystallinity of the CS structure and improve proton conductivity under hydrated conditions.

Various methods have been used to prepare phosphorylated chitin or CS. Jayakumar et al. [156] reported that phosphorylated CS can be prepared by heating CS with orthophosphoric acid and urea in *N*,*N*-dimethylformamide, because urea promotes the reaction [157]. The reaction of CS with phosphorous pentoxide (P_2_O_5_) in methanesulphonic acid was efficient, because the phosphorylated CS produced was water-soluble due to the high degree of substitution. Phosphorylated CS can also be prepared through the generation of N-methylene phosphonic CS by introducing methylene phosphonic groups into the CS matrix, thereby improving its solubility in water without affecting its filmogenic properties [158]. Jung et al. [159] reported on the preparation of phosphorylated CS through grafting mono (2-methacryloyl oxyethyl) acid phosphate to the CS matrix, thereby improving its zwitterionic and antimicrobial properties. Palma et al. [160] synthesized CS-*O*-ethyl phosphonate by using KOH/methanol and 2-chloro ethyl phosphonic acid under mild conditions; it was used to increase blueberry yields and soluble solid content. Phosphorylated CS can also be produced by using *n*-hexane as solvent rather than 2-propanol, because using the latter led to the production of undesired products. CS alkyl phosphate/CS-*O*-ethyl phosphonate has been synthesized under heterogeneous conditions; alkali CS, NaOH and *n*-hexane were used with diethyl chlorophosphate as phosphorylating agent. This phosphorylated CS was expected to enhance the reactivity of hydroxyl groups and favor the coupling reaction with diethyl chlorophosphate/2-chloroethyl phosphonic acid [161].

Wan et al. [162] prepared phosphorylated CS membranes through the reaction of orthophosphoric acid and urea in *N*,*N*-dimethylformamide; the phosphorylated CS membrane produced showed a decrease in the crystallinity structure due to increased phosphorus content. This phenomenon was due to the presence of bulky phosphate groups, which in turn increased the hydrophilicity and swelling properties of the membrane. Given that the crystallinity of the phosphorylated CS membrane decreased, the tensile strength also decreased. However, polyphosphate cross-linking did not affect the tensile strength and breaking elongation of the membranes. In phosphorylated CS membrane, additional –NH_2_ groups led to a high proton conductivity of 9.80 × 10^−4^ S cm^−1^ under hydrated conditions.

N-methylene phosphonic CS (NMPC) was produced by Dadhich et al. [163] by using microwave-assisted rapid synthesis through Mannich-type reaction (Figure 11). This reaction was developed, because the microwave irradiation can rapidly phosphorylate the CS with high and uniform-yield products without requiring a long reflux time, which is required in the phosphorylation method conducted by Moedritzer and Irani [164]. NMPC improved ionogenic propertyies, and the introduction of phosphonic or phosphonate groups into amino or hydroxyl groups improved biochemical function; thus, NMPC was selected as PEM for fuel cell applications [165,166]. When NMPC was synthesized through microwave irradiation for 10 min, the reaction yielded 55–60% of the products, in contrast with the low percentage of products (40–45%) of the reflux method for 14–24 h. Moreover, NMPC exhibited higher solubility properties than pure CS in aqueous medium. Proton conductivity improved with the increase in aqueous solution content to 5.0%. The highest proton conductivity of 10.15 mS cm^−1^ was obtained due to the existence of zwitterionic activity and proton transfer [167].

The phosphorylation of CS was also studied by Holder et al. [168]. They prepared phosphorylated-CS (CS-P) and phosphorylated-sorbitol-CS (S-CS-P) membranes. They claimed that phosphorylated CS membranes showed improved tensile strength and ion exchange capacity. Moreover, the fabricated PEM in microbial fuel cell (MFC) showed improved performance. The tensile strength of CS-P membrane was 49.9 N mm^−2^, which was higher compared with that of pure CS membrane (39.3 N mm^−2^) and S-CS-P membrane (0.4 N mm^−2^). The presence of sorbitol in S-CS-P membrane increased the flexibility of the CS structure, which promoted the decrement in the elongation and tensile strength of the membrane. Moreover, ion exchange capacity increased due to the introduction of high proton-conducting phosphate groups in the CS-P membrane, with the highest value of 1.43 mequiv g^−1^. The CS-P membrane reached a maximum power density of 130.03 mW m^−2^ and maximum current density of 374.73 mA m^−2^ when tested in the MFC system.

#### 2.3.3. Quaternization

Unmodified CS possessed low conductivity, i.e., 10^−9^ S cm^−1^ under anhydrous conditions and 10^−4^ S cm^−1^ under hydrated conditions. To improve this drawback, several researchers have modified CS through quaternization. The introduction of quaternary ammonium groups in CS matrix improves the solubility and positive charge density of CS [169,170,171,172], and the groups acted as positive charge-rich gels to be used as antimicrobial and antibacterial reagents [173,174]. Uragami et al. [175] reported that quaternized CS was synthesized using methyl iodide (Figure 12) [176] and by introducing the quaternizing agent glycidyltrimethylammonium chloride (Figure 13) [176], because it also had a quaternary ammonium group. A primary amino group at the C2 position of CS reacted with glycidyltrimethylammonium chloride [177]. The anion exchange conductor of quaternized CS was thus obtained by replacing hydroxide ions with chloride ions.

Wan et al. [178] fabricated novel cross-linked quaternized-CS membranes made from *N*-[(2-hydroxy-3-trimethylammonium) propyl] CS chloride (HTCC). The proton conductivities of these membranes were measured under anhydrous and hydrated conditions. The results showed increased proton conductivity of cross-linked quaternized-CS membranes with increasing degree of quaternization (DQ) of the HTCC. However, no clear trend was observed in the increment that occurred. Thus, the enhancement of cross-linked membrane conductance due to the HTCC with high DQ needs further study. The highest conductivity exhibited by the cross-linked membrane was 7.30 × 10^−3^ S cm^−1^ under hydrated conditions, which was near 10^−2^ S cm^−1^ and comparable with the conductivity of Nafion N117 membrane [179]. The single cell test was conducted at an H_2_/O_2_ pressure ratio of 241/241 kPa at 50 °C with 100% RH by using the cross-linked membrane with the highest proton conductivity due to its high DQ. The OCV achieved was approximately 1.0 V, and the current density was 65 mA cm^−2^.

Yuan et al. [180] proposed a new type of alkaline anion-exchange membrane by fabricating quaternized CS blended membrane by introducing 2,3-epoxypropyltrimethylammonium chloride (GTMAC) into CS matrix and produced *N*-[(2-hydroxy−3-trimethyl-ammonium) propyl] CS chloride (HTCC). GTMAC served an important function in the yield of conductive anions (OH^−^) as charge carriers, thereby avoiding process of quaternisation, which was toxic [178,181]. The quaternised CS blended membranes produced were *N*-[(2-hydroxy-3-trimethyl-ammonium) propyl] CS chloride (HTCC)/poly (diallyldimethylammonium chloride) (PDDA) (HTCC/PDDA-OH^−^) composite membranes. These membranes exhibited increased tensile strength with increasing PDDA content due to the function of PDDA as OH^−^ charge carriers and the cyclic structure of PDDA that inhibited the degradation of quaternary ammonium groups under alkaline conditions. The maximum tensile strength achieved by HTCC/PDDA-OH^−^ membrane with mass of 1:0.5 HTCC/PDDA was 25.31 MPa. However, as the ratio increased further, the membrane’s tensile strength decreased to 7.43 MPa. Moreover, the HTCC/PDDA-OH^−^ membrane also exhibited increased IEC and water uptake with maximum values of 3.267 mmol g^−1^ and 3.708 g g^−1^, respectively, due to the presence of additional quaternary ammonium groups and further enhancement in hydrophilicity properties when additional PDDA was introduced to the membrane. The proton conductivity of all membranes was measured by increasing the temperature from 30 to 80 °C. The HTCC/PDDA-OH- membrane with a mass of 1:0.5 showed the highest OH^−^ conductivity of 23.7 to 37.9 mS cm^−1^ from 30 to 80 °C. This enhancement in proton conductivity occurred because of the following: (i) the presence of water molecules that ensured the mobility of OH^−^ ions and ionic transport channel in the membrane; (ii) the activation energy for ion transport was overcome with increasing temperature, diffusion was rapid, and thermal motion of OH^−^ ions improved.

Quaternized CS (Q-CS) nanoparticles incorporated with quaternized poly (vinyl alcohol) (Q-PVA) polymer composite was prepared by Liao et al. [182]. The nanoparticles showed improved proton conductivity and fuel permeation. Lue et al. and Lue et al. [183,184] showed that as Q-CS nanoparticles were incorporated into Q-PVA, which had semicrystalline properties, the composite membrane showed enhanced conductivity and suppressed fuel permeation. The Q-CS nanoparticles acted as fillers in the membrane. Alkaline uptake was further increased from 0.56 g g^−1^ at 30 °C to 1.16 g g^−1^ at 30 to 60 °C. Proton conductivity values also increased from 0.020 to 0.022 S cm^−1^, with the lowest methanol permeability of 2.77 × 10^−6^ cm^2^ s^−1^. The Q-PVA/Q-CS composite membrane was operated in DMAFC and exhibited the highest peak power density of 92 mW cm^−2^ at 90 °C with 2 M methanol fuel feed.

Ryu et al. [185] showed that as quaternized CS underwent some modifications, producing a novel quaternized anion-exchange membrane, the conductivity and mechanical strength of the composite membrane have been improved relative to that of pristine CS membrane. The quaternized CS was modified with 4-pyridinecarboxaldehyde derivative, producing quaternized poly [O-(2-imidazolyethylene)-N-picolylchitosan (QPIENPC), which was then copolymerized with 1-vinylimidazole. The QPIENPC membrane exhibited increased proton conductivity from 3.48 to 10.15 mS cm^−1^ as temperature was increased from 30 to 80 °C. This value was higher than that of Q-CS (0.17 mS cm^−1^). The mechanical strength also improved in QPIENPC membrane, showing a higher tensile strength of 22.39 MPa relative to that of Q-CS (11.74 MPa). The single cell test was operated under alkaline conditions with 3 M methanol fuel feed and 2 M KOH solutions. The maximum peak power density obtained was 10.42 mW cm^−2^, thereby suggesting that the QPIENPC membrane was potentially suitable to be used as polymer electrolyte in DMAFC applications.

#### 2.3.4. Chemical Cross-Linking

Cross-linking is a common chemical modification method to ensure the good chemical and mechanical stability of CS and make CS insoluble in aqueous medium. CS membrane is cross-linked to reduce its high crystallinity properties and hence improve its water absorption and ionic transport through the membrane [186]. Cross-linking methods include chemical, physical, and oxidative cross-linking [187]. Structure and ionically cross-linked CS were reviewed by Berger et al. [188]. Polymer chains are interconnected by cross-linkers to form 3D networks in cross-linked CS. The main interactions forming the network are covalent or ionic bonds. In terms of their structure, covalently cross-linked CS can be divided into three types, as follows:
CS cross-linked with itself,Hybrid polymer networks, in which cross-linking reaction occurs between a structural unit of a CS chain and a structural unit of a polymeric chain of another type,Semi- or full-interpenetrating polymer networks, in which a polymer of another kind is entrapped in self-cross-linked CS network.


In addition to the ionic or covalent bonds, which are the main interactions that form networks, some secondary interactions, i.e., hydrogen bridges and hydrophobic interactions, also occur in CS networks. CS dissolved in weak organic acid solution becomes a polycation, which can form ionic cross-links with numerous cross-linking reagents, such a sulphuric acid, sulphosuccinic acid, acids of phosphate ions, and their salts. CS can also be cross-linked through covalent cross-linking process with reagents, such as glutaraldehyde (GA), formaldehyde, and epichlorohydrin.

CS cross-linking using sulphuric acid has been widely studied in for proton exchange and pervaporation, as shown in Figure 14 [189]. XRD of cross-linked CS samples shows that the intensity of semicrystalline peaks of CS membrane further decreases with increased duration of cross-linking with sulphuric acid. This occurrence indicated that as the duration of chitosan’s dissociation into the sulfuric acid increased, a high concentration of H^+^ ions was produced, which then occupied the SO_4_^2−^ ions, thereby proving that the SO_4_^2−^ ions were incorporated with NH_3_^+^ groups to form ionic bridges between CS polymer chains [189,190]. Improved ionic conduction occurs preferably in the amorphous phase, in which cross-linking ensures the reduced crystallinity phase of CS [186,191].

Ma et al. [192] prepared a cross-linked CS (CCS) membrane with sulphuric acid as cross-linking agent, which was then operated in direct borohydride fuel cell. The CCS membrane exhibits higher proton conductivity (1.10 × 10^−1^ S cm^−1^) when immersed in alkaline medium than in deionized water (6.20 × 10^−3^ S cm^−1^). This result is due to OH^−^ ions from the NaOH solution permeating the CCS membrane and forming hydrogen bonds with the addition of SO_4_^2−^ and Na^+^ ions. The single cell test conducted has shown that CCS membrane with CCH binder has better performance with the maximum peak power density of 450 mW cm^−2^ compared with CCS membrane with Nafion binder (402 mW cm^−2^) at 60 °C.

Vijayalekshmi and Khastgir [193] also studied the cross-linking of CS using sulphuric acid. They modified and produced the composite membrane by introducing methanesulphonic acid (MSA) and sodium salts of dodecylbenzene sulphonic acid (SDBS) into CS matrix and cross-linked the membrane with sulphuric acid. This modification was carried out to enhance the proton conductivity and mechanical strength of the composite membrane. Sulphuric acid was used as the cross-linking agent to improve proton conduction, and the SO3H groups from the MSA and SDBS also play important roles in providing additional charge hopping sites for proton transport in the membrane. The water uptake and ion-exchange capacity of CS-MSA and CS-SDBS membranes increased to 88.5% and 71.4% and to 0.35 and 0.27 mmol g^−1^ with increasing doping levels of MSA and SDBS, respectively. The mechanical strength and proton conductivity of CS-MSA and CS-SDBS membranes improved by approximately ≈35 MPa and 2.86 × 10^−4^ S cm^−1^ and 4.67 × 10^−4^ S cm^−1^ maximum at 100 °C, respectively, with increasing dopant level. This result proved that the usage of dopants and sulphuric acid as cross-linking agents helped improve mechanical strength and proton conductivity.

Pauliukaite et al. [194] stated that various dialdehydes, such as glyoxal and GA, are used to perform covalent cross-linking on –NH_2_ sites, thereby forming stable imine bonds between the amine groups of CS polymer and aldehyde groups. Figure 15 shows the cross-linking process of CS by using GA as cross-linking agent through Schiff’s base reaction, where an imine group is formed when the aldehyde group from the GA reacts with the amine group from CS matrix [186]. CS consists of both OH and NH_2_ groups. Thus, epichlorohydrin was preferably used as cross-linking agent, which reacted with OH groups under weak basic conditions for 2 h at 50 °C [195]. This technique retains the cationic amine groups, which helped promote ionic transport in membranes, as shown in Figure 16 [176].

Dashtimoghadam et al. [196] studied the cross-linking of CS with binary cross-linking agents, which were mixtures of sulphosuccinic acid (SSA) and GA to form CS network membranes, with various compositions of cross-linking agents. With increased SSA loadings, the proton conductivity of cross-linked CS membrane also increased, with the highest value of 0.0452 S cm^−1^ at 16 wt.% SSA. This increment was due to the formation of additional pathways for proton migrations because of the presence of –SO_3_ groups occupying the membrane. The methanol permeability and selectivity achieved for this membrane were 9.60 × 10^−7^ cm^2^ s^−1^ and 47,100 S cm^−3^ s, respectively. The single-cell performance test was conducted in DMFC, and the maximum power density obtained was 17 mW cm^−2^ at 30 °C and 41 mW cm^−2^ at 60 °C with 2 M methanol fuel feed. Thus, the modified CS biopolymer is a potential polyelectrolyte membrane for green power generation fields.

Hasani-Sadrabadi et al. [197] fabricated a novel triple-layer proton exchange membrane consisting of two layers of modified CS coated with Nafion on both sides, which acted as methanol barrier layers. The modified CS layers were also cross-linked with binary cross-linking agents of SSA and GA. The proton conductivity of the triple-layer membrane (CGS−12/Nafion 105/CGS-12) increased as the temperature increased from 25 to 90 °C, with values of 0.088 to 0.1635 S cm^−1^, respectively. The methanol permeability of the triple-layer membrane increased with increasing temperature from 25 to 80 °C (2.52 × 10^−7^ to 7.74 × 10^−7^ cm^2^ s^−1^). However, these values can be considered much lower than the methanol permeability of Nafion 105 membrane (3.01 × 10^−6^ cm^2^ s^−1^). The triple-layer membrane was operated in DMFC and exhibited the highest peak power density of 68.10 mW cm^−2^ at 70 °C with 5 M methanol fuel feed.

The cross-linking of modified CS membrane by using the binary cross-linking agents SSA and GA was studied by Srinophakun et al. [198]. This membrane has been applied as PEM in MFC. SSA and 3-chloro-2-hydroxypropyl trimethylammonium chloride (Quat-188), which acted as proton carrier sites and positively charged molecules, have been introduced into CS polymer matrix and produced quaternized CS membrane. The water uptake, proton conductivity, and ultimate tensile strength of the modified CS membrane with 0.6 mole ratio of SSA exhibited the highest values of ≈130%, 0.0554 mS cm^−1^, and 6.698 MPa, respectively, with increasing SSA content. This membrane with 0.6 mole ratio of SSA was quaternized with Quat-188 for 8 h as a further step and has resulted in high proton conductivity (0.9919 mS cm^−1^). The ultimate tensile strength was decreased to 2.62 MPa as it was quaternized for 4 h due to its fragility. Hence, using only SSA and GA as cross-linking agents helped improve the proton conductivity and mechanical strength of the modified CS membrane.

As conclusion, various methods of modifications on CS-based membranes have been performed by researchers, including sulfonation, phosphorylation, quaternization, and cross-linking methods. Each of the methods have shown their ability to improve the efficiency and fuel cell performance of the membranes. CS composite membranes that have been modified using the sulfonation method, in which the CS polymer is functionalized with sulfonic groups exhibited enhancement in proton conductivity, due to the proton hopping sites available for proton transfer to occur, without improving their solubility. Meanwhile, as the CS membranes are modified through phosphorylation and quaternization methods, it has been reported by previous research that both phosphonic groups and quaternary ammonium groups in each of the methods, respectively, have the ability to enhance the solubility of CS membranes, reduce their crystallinity, increase their hydrophilicity properties, and hence enhance the proton conductivity of the membranes as there is the presence of zwitterionic activity and proton transfer in the membranes. Other than that, the CS composite membranes can be further improved when they are cross-linked with cross-linking agents such as sulphuric acid, GA, SSA, etc. However, according to previous works, as the CS composite membranes were cross-linked with binary cross-linking agents (GA and SSA), the achieved proton conductivity was much higher than the individual cross-linking agent, due to the supplementary proton-conducting pathway developed for proton migrations in the membrane. All of these methods have also proven to improve the mechanical strength of the membranes and, based on previous reported works, it can be observed that the most promising method of improving the membranes is by functionalizing the CS polymer and consequently cross-linking the CS composite membranes, to be used as PEM in fuel cell applications.

## 3. Ionic Liquids in Polymer Electrolyte Membrane Fuel Cell

### 3.1. Proton-Conducting Ionic Liquids (PCILs)

Various approaches of improving and enhancing stable polymer electrolyte membranes with good conductivity and water retention capacity have been proposed, including acid–base composites, incorporation of nanofillers into polymer matrix, chemical cross-linking, and dispersion of ceramic fillers or functionalization of polymer side chains. However, the conductivity values obtained (10^−4^ to 10^−3^ S cm^−1^) were insufficient and poor under anhydrous conditions, as the proton hopping sites on GO were ineffective, according to Zhao et al. [199]. Ionic liquids have been introduced into polymer electrolyte membranes to promote the anhydrous conductivity to reach a significant level, which was approximately 10^−2^ S cm^−1^ [200]. The incorporation of ionic liquids into polymer matrix was utilized via two methods. The first method involved adding ionic liquids directly into the casting solution or by impregnating them into polymer membrane matrix [201,202,203,204]. The second involved attaching or embedding the ionic liquids on a filler and then introducing them into the polymer matrix or grafting them directly onto the polymer [205,206,207].

Ionic liquids are molten salts with melting points below 100 °C or room temperature and are therefore known as room temperature ionic liquids. Ionic liquids are powerful solvents and electrolytes that conduct electricity. In PEMFC, Nafion polymer is used as a common electrolyte material for ionic conduction; however, its efficiency is only approximately 40% [208]. This percentage can be increased if new materials are added to the electrolyte used in PEMFC. The addition of ionic liquids can promote proton conductivity in PEMFC, as they are good electrolytes.

Ionic liquids have been widely used due to their low vapor pressure; low melting point; high ionic conductivity; high thermal, chemical and electrochemical stability and nonvolatility; hence, they are extensively studied for high-temperature fuel cells [209]. Ionic liquids are completely formed by ions and from organic cations and organic or inorganic anions as they do not contain a solvent. Organic cations, such as piperidinium, pyrrolidinium, imidazolium, pyridinium, tetraalkylammonium, tetraalkylphosphonium, and sulphonium, are typically used. The anions used in preparation of ionic liquids include BF_4_, PF_6_, Tf_2_N, TfO, and chloride.

Ionic liquids occur in two types, namely, protic and aprotic. Protic ionic liquids are formed by stoichiometric neutralization reactions of a Brønsted acid (proton donor) and Brønsted base (proton acceptor) and have exchangeable protons (active protons). The first reported ionic liquid is the protic ionic liquid ethanolammonium nitrate, followed by ethylammonium nitrate. Protic ionic liquids are different from other ionic liquids because of the presence of available protons that can build up extensive hydrogen bonding and allow unique applications. However, despite extensive studies on the physicochemical properties of protic ionic liquids, the bulk phase structure of the materials remains inadequately understood [210,211,212,213,214]. Proton migration occurs in protic ionic liquids through vehicular mechanism, where they have highest conductivities and possess highest fluidities [215]. The advantage of using protic ionic liquids is that PEMFC can be performed at temperatures above 100 °C. Furthermore, they can overcome volatile electrolyte problems under anhydrous conditions, because proton transport is independent of water content. Aprotic ionic liquids are synthesized by the quaternization (alkylation) of the corresponding amine or imidazole, followed by an anion exchange reaction. These types of ionic liquids have no exchangeable protons (no active protons) in their chemical structure and are thus called aprotic ionic liquids [216].

Ionic liquid conductivities at room temperature range from 1.0 × 10^−4^ to 1.8 × 10^−2^ S cm^−1^. For ionic liquids based on dialkyl-substituted imidazolium cations, the conductivities are generally 1.0 × 10^−2^ S cm^−1^. However, the conductivities of ionic liquids based on tetraalkylammonium, pyrrolidinium, piperidinium, and pyridinium cations are lower, ranging from 1.0 × 10^−4^ to 5.0 × 10^−3^ S cm^−1^. Cation reduction potential and anion oxidation depend on counter ions. Thus, halide anions, such as F^−^ and Br^−^, prohibit the stability to 2.0–3.0 V. Furthermore, bis(trifluoromethylsulphonyl)imide anions and Tf_2_N- are oxidized at a high anodic potential, thereby allowing stability at 4.5 V [217]. Ionic liquids based on tetraalkylammonium cations are reduced at moderately negative cathodic potentials, which are characterized by an improved stability of 4.0–5.7 V [218].

Large aggregates in the form of micelles or lamellae [212] form in ionic liquids through dipole-dipole interactions or hydrogen bonding. The aggregation behavior depends mainly on the nature of cations and anions that are chemically present in PCILs. Monoalkylammonium-based ionic liquids show the formation of large aggregates regardless of the alkyl length chain, whereas di- and tri-alkylammonium-based ionic liquids show a low tendency to form stable aggregates as a result of their high steric hindrances and low amount of hydrogen present for hydrogen bonding. Furthermore, the tendency to form aggregates can cause decreased ionic conductivity in PEMFC [219].

The physical and chemical properties of ionic liquids can be enhanced by mixing them with organic and inorganic compounds [220]. Inorganic glasses have excellent chemical and thermal stability, high mechanical strength, and low cost. Li et al. [221] developed hybrid proton exchange membranes based on diethylmethylammonium trifluoromethanesulphonate ([dema][TfO]) and SiO_2_ monoliths, which exhibited very high anhydrous ionic conductivities exceeding 1.0 × 10^−2^ S cm^−1^ at 120–220 °C.

Luo et al. [222] promoted the comparison of physicochemical properties of phosphonium and ammonium protic ionic liquids with trioctyl and triphenyl groups. Phosphonium-based protic ionic liquids show higher thermal stability than ammonium-based protic ionic liquids. In both phosphonium and ammonium ionic liquids, those with octyl group exhibited high thermal stability. Phosphonium ionic liquids exhibit higher conductivity than ammonium ionic liquids because of their weaker hydrogen bond, Coulombic interactions, and higher carrier ion concentrations [223].

### 3.2. Modification of CS in Ionic Liquids

The modification of CS in ionic liquids through homogeneous reaction has drawn much attention. Wei et al. [224] prepared CS-ionic liquid grafted polymer (CS-IL) in 1-carboxymethyl –butyl−3-methylimidazolium chloride ([Cmim]Cl) through homogenous reaction between carboxyl and –NH_2_ at 100 °C. The CS-IL has excellent adsorption capacity for Cr_2_O_7_^2−^ (0.422 mmol/g) and PF_6_^−^ (0.840 mmol/g) anions. The adsorption capacity increased with the grafted degree of [Cmim]Cl and with decreasing aqueous pH. This result was ascribed to the protonation of –NH_2_ at low pH, which interacted with anions through electrostatic force. The CS-IL adsorbed anion aggregates in aqueous solution and precipitates from bulk solution for separation.

Wang et al. [225] prepared polycaprolactone-grafted CS (CS-g-PCL) in [Emim]Cl through reaction between polycaprolactone and –NH_2_ in CS with stannous octoate as catalyst at 100 °C. The structure and grafting degree of CS-g-PCL can be modulated by changing the reaction conditions and reactant material ratio. The glass transition temperature (T_g_) of CS-g-PCL was lower than that of CS due to the change of polymer crystallinity and polymer composition. The T_g_ of PCL was approximately −60 °C. The technique provided a novel method for the preparation of CS-g-PCL for use in tissue engineering, drug and gene transfer.

Xu et al. [226] prepared two kinds of monomethyl fumaric acid (MFA)-modified CS (MF-CS) with different chemical structures in ionic liquids. One was an MF-CS salt composed of CS cation and MFA anion synthesized in a mixture of 1-ethyl-3-(3-dimethylaminopropyl) carbodiimide (EDC) and N-hydroxysuccinimide. In MF-CS molecules, MFA and CS were covalently attached. The other type was an MF-CS amide synthesized in EDC. The antioxidant activity of the two CS derivatives improved due to the introduction of MFA.

### 3.3. CS-Ionic Liquid Composite Membrane

The addition of ionic liquids in membrane matrix can help enhance the membranes′ performance. Thus, a suitable polymer matrix host must be determined. Biopolymers were preferred for use as the polymer host, because they were low cost, nontoxic, biodegradable, and can be extracted from renewable sources. CS is favored and widely used in the fabrication of composite membranes due to its abundance and attractive properties. It has also been combined with various fillers and proton conductors. However, the conductivity and performance achieved is insufficient, and its applications remained limited, because it is composed of low-molecular-weight CS [6,227,228]. Thus, CS is often selected as the polymer matrix for combination with ionic liquids to improve the performance of the composite membrane produced.

Singh et al. [229] developed and characterized 1-ethyl-3-methylimidazolium thiocyanate ([Emim]SCN) ionic liquid doped with biopolymer composite membrane based on CS, which showed improved ionic conductivity relative to that of pure, unmodified CS membrane due to the enhancement in the number of charge carriers provided by the ionic liquid. The composite membranes were prepared using different compositions of ionic liquids (10–250 wt.%) in CS solution and mixed thoroughly. The ionic conductivity increased with increasing ionic liquid composition, attaining maximum values, and then decreased. The maximum ionic conductivity was 2.60 × 10^−4^ S cm^−1^ at 150 wt.% ionic liquid composition, because this ionic conductivity value was quite high compared with the value of ionic conductivity of pure CS. The enhancement of ionic conductivity was influenced by the doping of ionic liquids, which provided additional free ions (imidazolium cation and thiocyanate anions) and increased the conductivity. The decreased conductivity was due to the aggregation of charge carriers [229]. Based on the XRD diffraction measurement, the intensity of crystalline region further decreased with increasing ionic liquid content in the membrane matrix, thereby increasing the amorphous region and proving the increment in ionic conductivity.

Anhydrous conductive CS membranes have been prepared by combining CS with functional ionic liquids. Xiong et al. [230] synthesized 1,4-*bis*(3-carboxymethyl-imidazolium)-1-yl butane ([CBIm]) ionic liquid, which consisted of different anions, namely, chloride, acetate, tetrafluoroborate, and iodide. CS was dissolved in water in the presence of these ionic liquids to produce an anhydrous conductive membrane (Figure 17). The composite membranes based on CS-ILs were then prepared by casting.

At room temperature under anhydrous conditions, the conductivities were enhanced to approximately 10^−4^ S cm^−1^. Among these CS-IL composite membranes with different anions, including Cl^−^, Ac^−^, BF_4_^−^, and I^−^, CS-[CBIm]Cl composite membranes exhibited the highest ionic conductivity value of 0.68 × 10^−4^ S cm^−1^. The small size of Cl^−^ presented the highest mobility compared with other anions, and [CBIm]Cl provided more H^+^ at the same amount of incorporated IL, because its molecular weight was the smallest [230]. Moreover, the conductivity of composite materials can be enhanced through doping [231,232]. In this study, CS-[CBIm]I composite membranes were doped with different amounts of iodine. The I_3_-/I^−^ redox couple could be produced due to the existence of I^−^, when KI and I_2_ were added into the composite materials that could considerably improve conductivity. The conductivity of composite materials depends on the number of I_3_^−^ and I^−^ ions and their ability to perform the redox reaction 3I^−^ ↔ I_3_^−^ + 2e^−^ [233]. Given that CS-[CBIm]I composite membrane was doped with iodine, the conductivity increased as expected. At room temperature, conductivity reached a level as high as 0.91 × 10^−2^ S cm^−1^ when doped with 0.25 mmol iodine. This conductivity was 200 times that of the undoped composite membrane. However, its conductivity decreased quickly when it was doped with more than 0.25 mmol iodine; the excessive iodine amount breaks the redox equilibrium of I_3_^−^ and I^−^ and decreases the transfer efficiency [230].

Leones et al. [234] stated that ionic liquids are preferred over common inorganic salts and are compatible for mixing with most polymer matrices, thereby allowing the development of composite membranes with smooth structure and good flexibility. This statement was proven true in the study conducted by Shamsudin et al. [235,236], which involved a mixture of chitosan matrix with 1-butyl-3-methylimidazolium acetate [Bmim][OAc]. From SEM micrographs (Figure 18), it was observed that with increasing addition of [Bmim][OAc], the surface morphology was transformed and became smoother with smaller pores, thereby indicating the reduction of crystalline phase and enhancement of amorphous region in the polymer matrix. The highest ionic conductivity achieved at 90 wt.% of [Bmim][OAc] was 2.44 × 10^−3^ S cm^−1^ at room temperature, which was higher than that previously observed [229]. Thus, [Bmim][OAc] is more compatible, shows better charge carrier for CS, and forms better interaction with the polymer matrix compared with others. Ionic conductivity increased with increasing [Bmim][OAc] content, and amorphous phase of the polymer chain was enhanced. Thus, the inter- and intra-molecular hydrogen bonds of CS was diminished by the ionic liquids, hence extinguishing the crystallinity of the membrane.

### 3.4. Proton Transport Mechanism through Ionic Liquids

Proton conductivity is a necessary feature needed in evaluating the performance of membrane in fuel cell applications. Two types of proton conduction mechanisms are proposed, namely, Grotthuss and vehicle mechanisms (Figure 19) [216]. Grotthuss mechanism, which is also known as proton jumping, involves protons hopping through the formation and breaking of hydrogen bonding network. Vehicle mechanism involves the diffusion of protons with the support of moving ‘vehicle’. Protons movement can occur rapidly through the Grotthuss mechanism, which was in contrast to the vehicle mechanism in the existence of strong hydrogen bonding networks [216,237].

Noda et al. [238] studied a mixture of solid bis-(trifluoromethanesulfonyl)amide (HTFSI) and solid imidazole (Im) with various molar ratios. According to Kreuer et al. [239], intermolecular proton transfer is determined by proton transport between the protonated Im (HIm^+^) and neat Im, namely, proton defects, at an optimal composition, thereby enhancing the proton conduction. Proton transfer is stimulated particularly by HIm^+^ species in neat Im through Grotthuss mechanism in Im excess content. The achieved conductivity increases with increasing Im mole fraction. This result is due to improved ionic mobility, which is related to proton conduction occurring through the combination of Grotthuss-and vehicle-type mechanisms. The prominent conducting properties are then changed from vehicle to Grotthuss mechanism, because the Im mole fraction is increased from the equimolar salt.

Ye et al. [240] prepared phosphoric acid/1-methyl-3-propyl-methylimidazolium dihydrogen phosphate/polybenzimidazole (H_3_PO_4_/PMIH_2_PO_4_/PBI) composite membrane, which was a 3D hydrogen bonded network complex that intensified the structural stability of the membrane and the acid/ionic liquid stability. The proton transport of this complex membrane occurs through hydrogen bonding network by Grotthuss mechanism (Figure 20). This occurrence was due to the enhancement of proton conductivity caused by the large number of defect protons, i.e., H_2_PO_4_^−^ ions from the ionic liquid, which served as proton exchange sites [241]. Good proton transfer reaction requires strong hydrogen bonding, whereas weak hydrogen bonding promotes bond breakage and reformation to accomplish long-range proton transport [242].

## 4. Conclusions and Future Perspectives

Various studies have been performed to prove that CS-based biopolymer membrane is suitable for use in various fuel cell applications (PEMFC, DMFC, MFC, DMAFC, and others) because of its low-cost, eco-compatibility, and thermal and chemical stability properties. Despite these advantages, it has low proton conductivity and low mechanical strength. Modifications in the CS matrix have been made, including addition of functionalized inorganic fillers (hygroscopic oxides, heteropoly acids, carbon nanotubes, and graphene oxide), CS/polymer blends, and chemical modifications (sulphonation, phosphorylation, quaternization, and chemical cross-linking) to enhance this property. CS membrane can also be improved by adding other proton conductors, which are ionic liquids that can promote proton conductivity in PEMFC, as they are recognised as good electrolytes. The performance of mixtures of ionic liquids and polymers have been widely explored to enhance the physical and chemical properties of the composite membranes.

Modifications in CS membranes are widely studied to improve various properties, such as excellent proton conductivity, high mechanical and thermal stabilities, low methanol permeability, and good performance in fuel cell applications. However, the modifications performed on CS still have some limitations that must be addressed to produce good membranes and to be used as PEMs in economical and good-quality fuel cells. CS membranes modified by introducing heteropoly acids and ionic liquids can encourage the occurrence of leakage due to their solubility properties. Given the high hydrophilicity and excessive water uptake of some modified CS membranes, their mechanical properties are often hindered, leading to incompatibility in their use as PEMs in fuel cell applications.

In future studies, several improvements must be made to CS composite membrane to produce improved performance for fuel cell applications. The features that must be improved include the following:
Mechanical stability. A composite membrane with improved mechanical strength should be produced to ensure its sustainability in long-duration fuel cell operations.Water retention capacity. Composite membranes with good retention capacity must be developed to prevent the loss of bound water when tested at high temperatures.Proton conductivity. Biopolymer composite membranes with increased conductivity and that are comparable with Nafion membranes must be developed.Low cost. Inexpensive composite membranes that are environmentally friendly and have attractive properties must be developed.


## Figures and Tables

**Figure 1 ijms-21-00632-f001:**
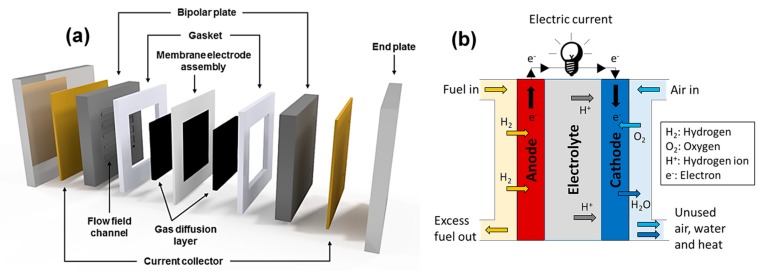
(**a**) Main components of a polymer electrolyte membrane fuel cell (PEMFC) and (**b**) schematic of a typical membrane electrode assembly (MEA).

**Figure 2 ijms-21-00632-f002:**
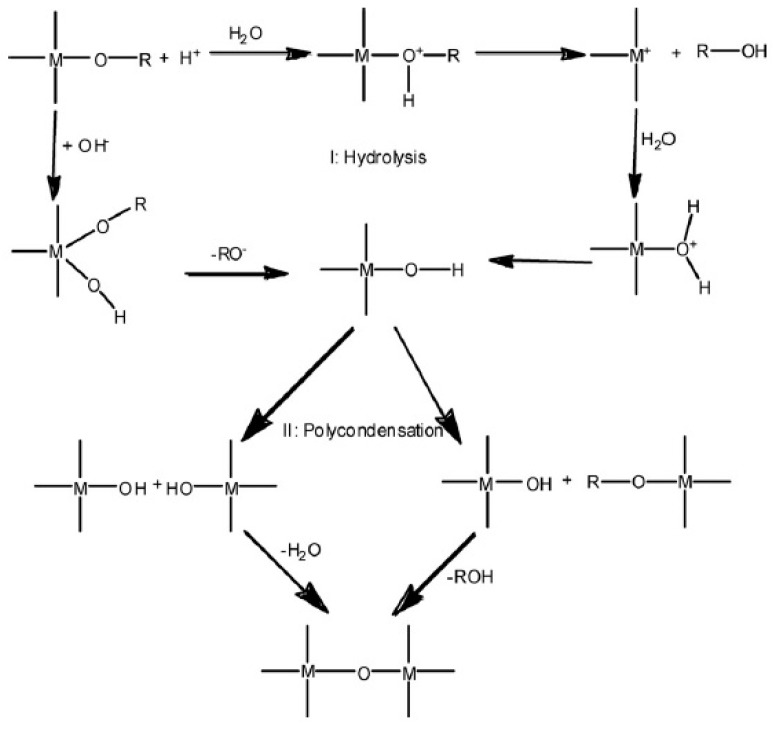
Schematic of the two-step sol–gel reaction routes for metal alkoxides [16]. Copyright 2011, reproduced with permission from Elsevier

**Figure 3 ijms-21-00632-f003:**
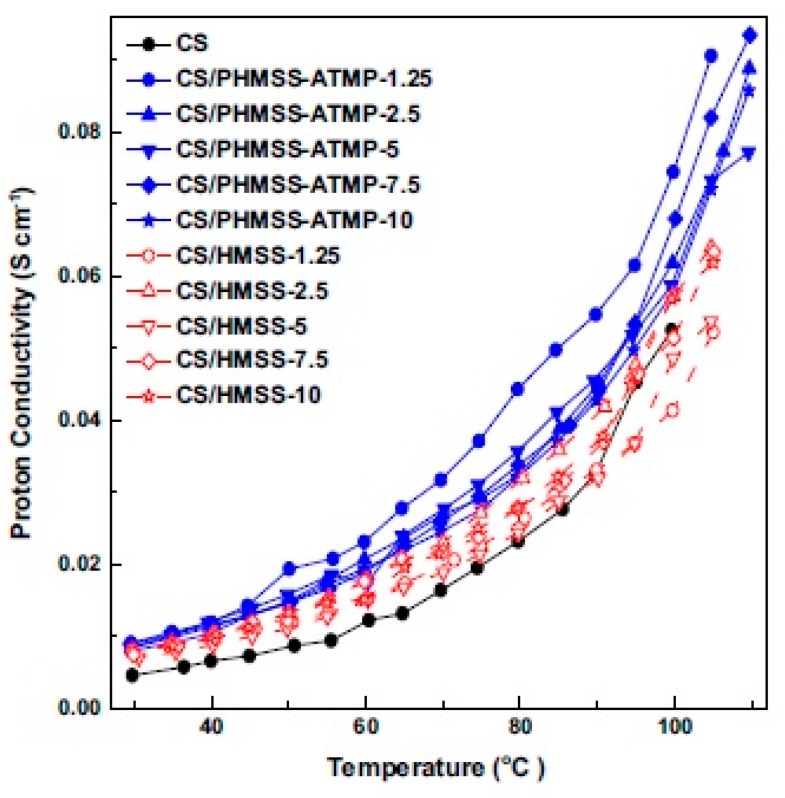
Proton conductivity of the pristine chitosan (CS), CS/HMSS, and CS/phosphorylated hollow mesoporous silica sub-microspheres (PHMSS)-ATMP hybrid membranes with increasing temperature at 100% relative humidity (RH) [28]. Copyright 2014, reproduced with permission from Elsevier

**Figure 4 ijms-21-00632-f004:**
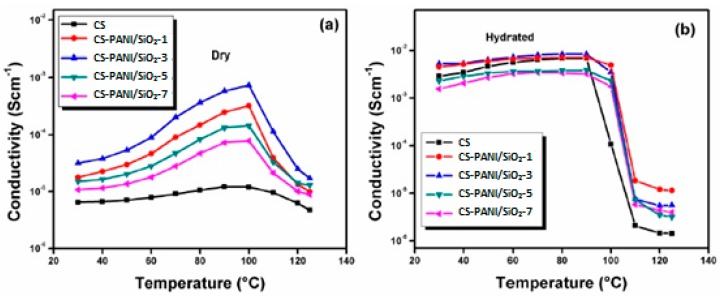
Proton conductivity of (**a**) dry and (**b**) hydrated CS-polyaniline/nanosilica (PANI/SiO_2_) hybrid membranes with increasing temperatures [26]. Copyright 2018, reproduced with permission from Elsevier

**Figure 5 ijms-21-00632-f005:**
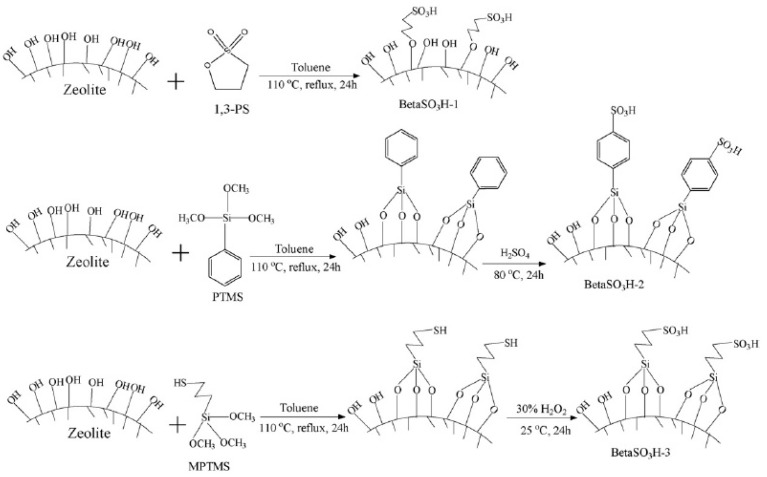
Schematic of three different pathways for the sulphonation of zeolite beta particles [17]. Copyright 2008, reproduced with permission from Elsevier

**Figure 6 ijms-21-00632-f006:**
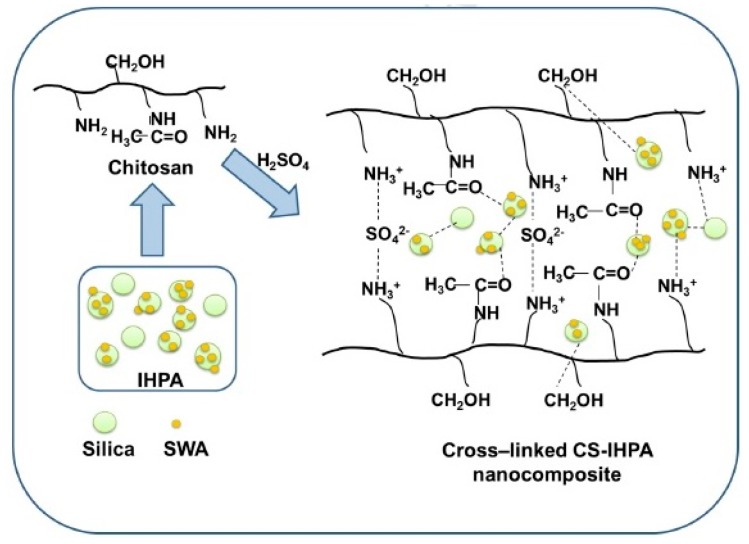
Schematic of the structure of crosslinked CS-silica-supported silicotungstic acid (IHPA) nanocomposite membranes [60]. Copyright 2018, reproduced with permission from Elsevier

**Figure 7 ijms-21-00632-f007:**
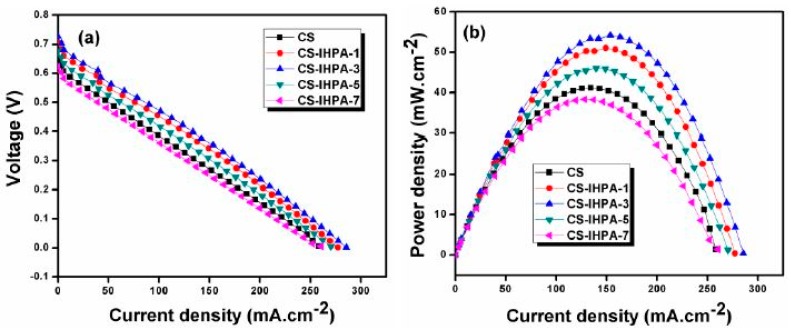
(**a**) Polarization and (**b**) power density curves of CS-IHPA nanocomposite membranes operated in DMFC at 80 °C [60]. Copyright 2018, reproduced with permission from Elsevier

**Figure 8 ijms-21-00632-f008:**
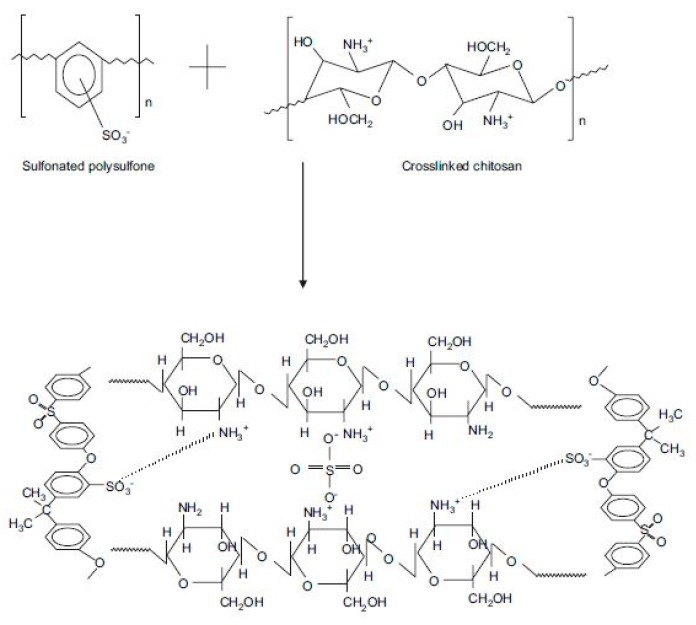
Schematic of CS and sulphonated polysulphone (PSF) blend membrane [127]. Copyright 2008, reproduced with permission from Elsevier.

**Figure 9 ijms-21-00632-f009:**
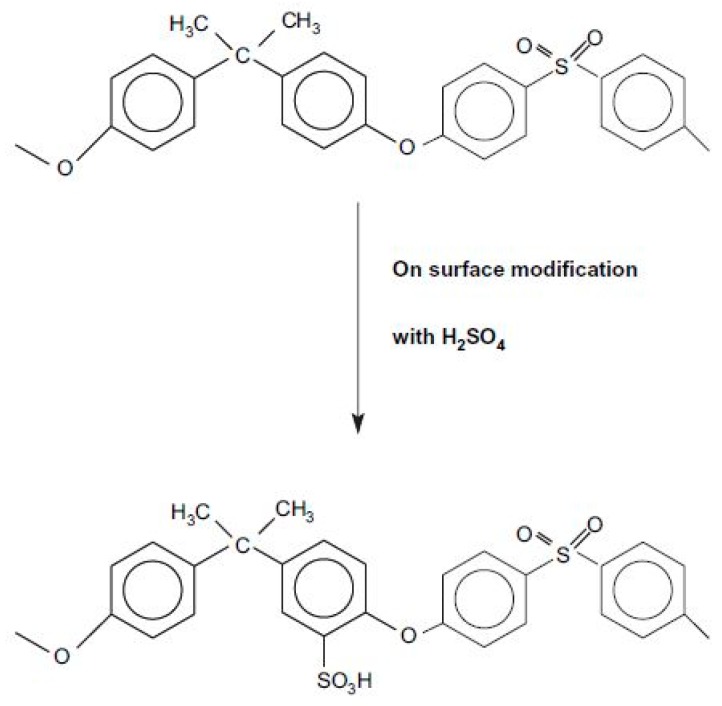
Schematic of the surface modification of PSF through sulphonation [127]. Copyright 2008, reproduced with permission from Elsevier.

**Figure 10 ijms-21-00632-f010:**
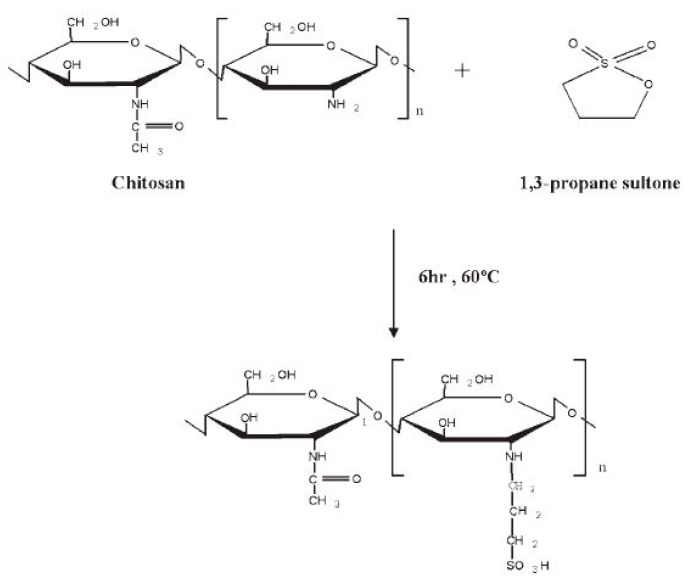
Schematic of the structure of sulphonated CS [152]. Copyright 2010, reproduced with permission from John Wiley and Sons.

**Figure 11 ijms-21-00632-f011:**
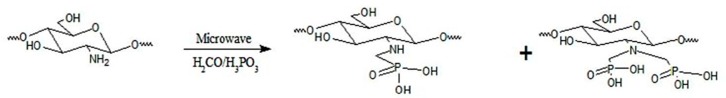
Synthesis of N-methylene phosphonic CS via microwave irradiation reaction [163]. Copyright 2015, reproduced with permission from Elsevier

**Figure 12 ijms-21-00632-f012:**
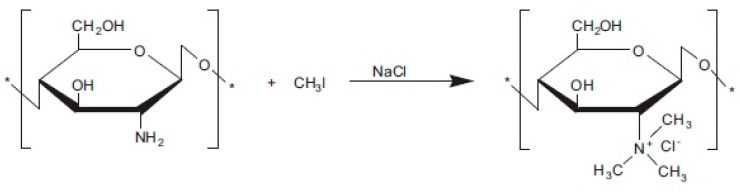
Synthesis of quaternized CS [176]. Copyright 2013, reproduced with permission from Elsevier

**Figure 13 ijms-21-00632-f013:**
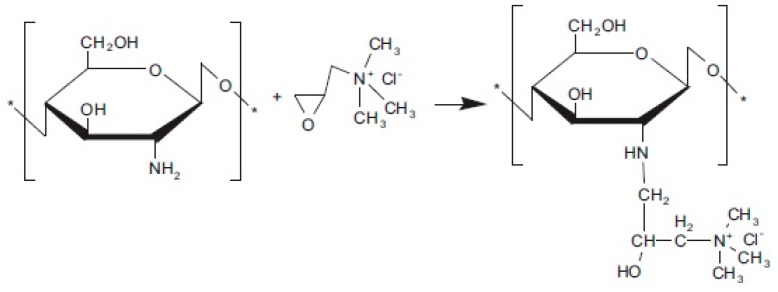
Reaction of CS with glycidyltrimethylammonium chloride [176]. Copyright 2013, reproduced with permission from Elsevier

**Figure 14 ijms-21-00632-f014:**
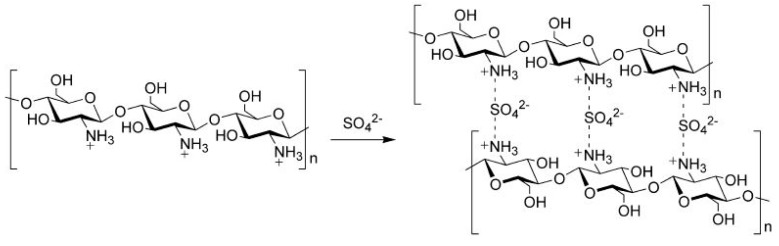
Chemical structure of CS cross-linked with sulphuric acid [189]. Copyright 2008, reproduced with permission from Elsevier

**Figure 15 ijms-21-00632-f015:**
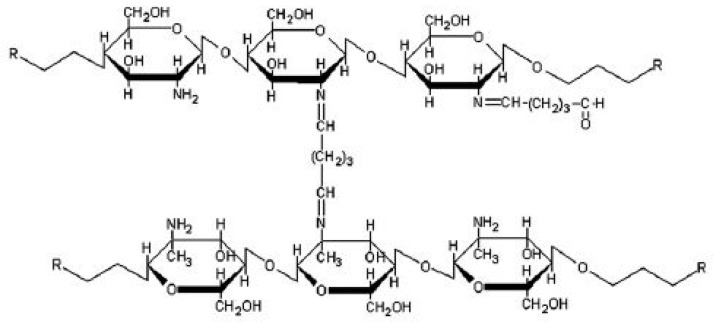
Structure of CS cross-linked with glutaraldehyde (GA) through Schiff’s base reaction [186]. Copyright 2010, reproduced with permission from Elsevier

**Figure 16 ijms-21-00632-f016:**
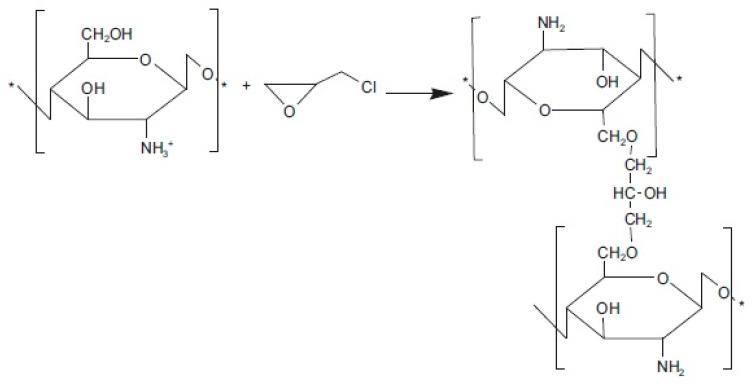
Cross-linking process of CS treated with epichlorohydrin [176]. Copyright 2013, reproduced with permission from Elsevier

**Figure 17 ijms-21-00632-f017:**
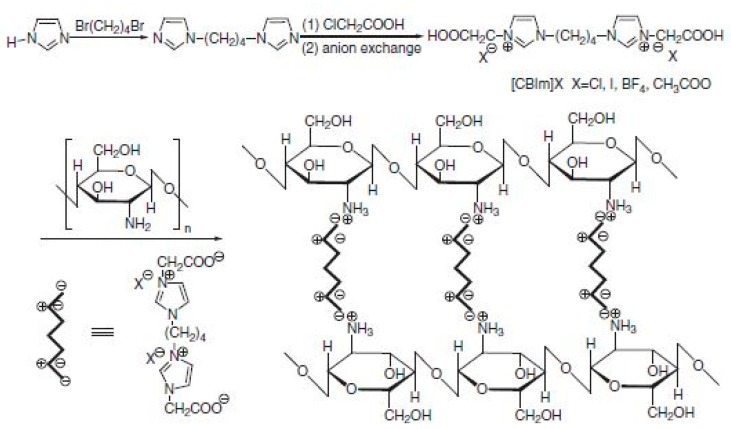
Schematic of the structure of CS-[CBIm]X composite membranes [230]. Copyright 2011, reproduced with permission from John Wiley and Sons

**Figure 18 ijms-21-00632-f018:**
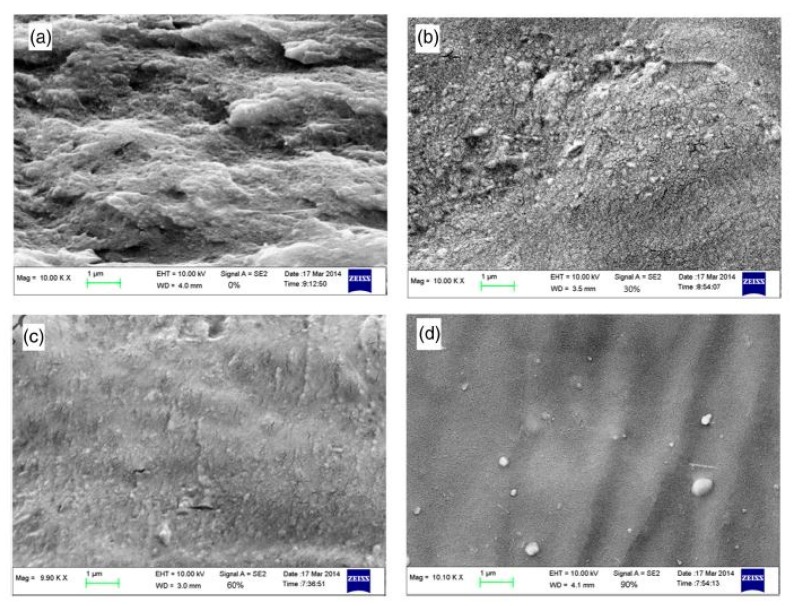
SEM micrographs of CS membrane with the addition of (**a**) 0 wt.%, (**b**) 30 wt.%, (**c**) 60 wt.%, and (**d**) 90 wt.% [Bmim][OAc] [236]. Copyright 2015, reproduced with permission from Elsevier

**Figure 19 ijms-21-00632-f019:**
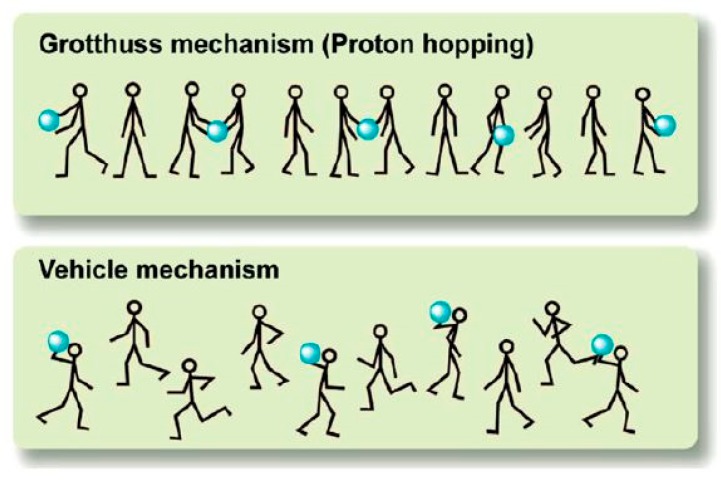
Illustrations of proton conduction mechanisms in acidic aqueous solutions and protic ionic liquids [216]. Copyright 2013, reproduced with permission from Cambridge University Press

**Figure 20 ijms-21-00632-f020:**
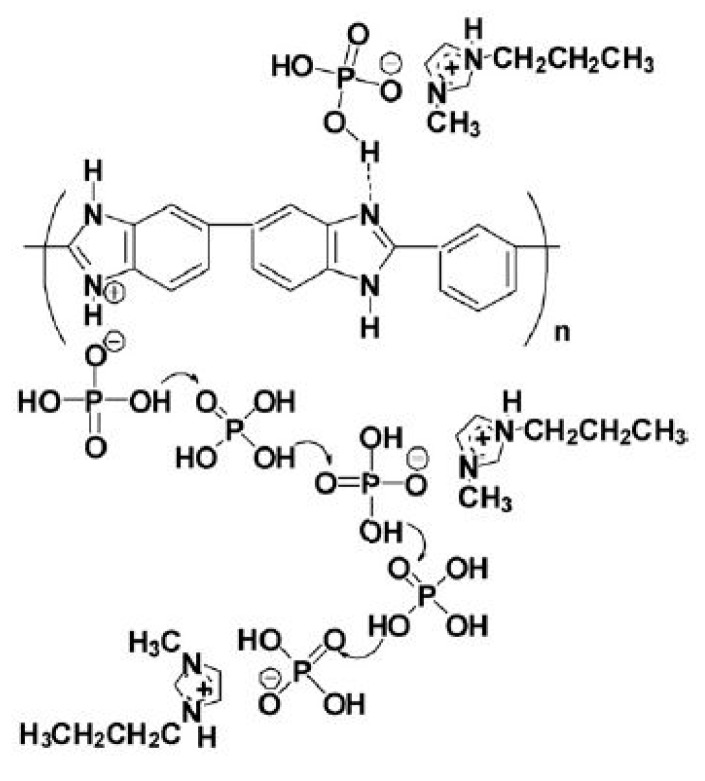
Schematic of the structure of proton transport mechanism in the H_3_PO_4_/PMIH_2_PO_4_/PBI membrane under anhydrous condition [240]. Copyright 2008, reproduced with permission from Elsevier

**Table 1 ijms-21-00632-t001:** Applications of hygroscopic oxide fillers in CS hybrid membranes.

**Membrane**	**Inorganic Fillers**	**Synthesis Method**	**Performance**	**Application**	**References**
CS/sulphonated graphene oxide/silica (CS/sGO/SiO_2_)	Sulphonated graphene oxide/silica (sGO/SiO_2_)	Incorporation of sGO/SiO_2_ into the CS matrix Solution-casting method membrane	Improved water uptake (49.10–59.30%) Reduced ion exchange capacity (0.29–0.23 mmol g^−1^) Increased proton conductivity (0.068–0.092 S cm^−1^) Max power density: 36.31 mW cm^−2^ at 30 °C Max power density: 87.18 mW cm^−2^ at 80 °C	DMFC	[39]
CS sulphate	Nanosilica	Nanosilica synthesis using precipitation method Solution-casting method membrane	Increased water uptake (40.75%) Increased ion exchange capacity (0.66 mequiv g^−1^) Highest proton conductivity: 6.48 × 10^−4^ S cm^−1^ Increased methanol absorption (44.22%)	DMFC	[40]
Sulphonated CS/poly (ethylene oxide)/sulphonated silica (s-CS/PEO/s-SiO_2_)	Sulphonated silica (s-SiO_2_)	Sulphonation of CS and silica Solution-casting method membrane	Increased water uptake (47.00–55.00%) Improved ion exchange capacity (0.73–0.91 mequiv g^−1^) Highest proton conductivity: 4.76 × 10^−2^ S cm^−1^ at 25 °C Improved tensile strength (10.60–17.63 MPa)	PEMFC	[41]
CS-polyaniline/nano silica (CS-PAni/SiO_2_)	Polyaniline/nanosilica (PAni/SiO_2_)	PAni/SiO_2_ synthesis using a facile ultrasonication method Solution-casting method membrane	Improved tensile strength (65.1–101.5 MPa) Increased ion exchange capacity (0.35–0.96 mmol g^−1^) High water uptake (57.62–72.87%) Highest proton conductivity: 8.39 × 10^−3^ S cm^−1^ at 80 °C Max power density: 56.27 mW cm^−2^ at 80 °C	DMFC	[26]
CS/zwitterion functionalized titania-silica (CS/TiC-SiN)	Zwitterion functionalized titania-silica (TiC-SiN)	TiC–SiN synthesis using facile chelation method Solution-casting method membrane	Increased water uptake Increased methanol permeability Improved ion exchange capacity (0.356–0.587 mmol g^−1^) Highest proton conductivity: 0.0408 S cm^−1^ at 25 °C Highest selectivity: 4.85 × 10^4^ S cm^−3^ s	DMFC	[31]
CS/phosphorylated hollow mesoporous silica sub-microspheres-amino tris (methylene phosphonic acid) (CS/PHMSS-ATMP)	Phosphorylated hollow mesoporous silica sub-microspheres-amino tris (methylene phosphonic acid) (PHMSS-ATMP)	Phosphorylation of hollow mesoporous silica sub-microspheres Solution-casting method membrane	Increased water uptake (47.8%) Reduced swelling properties (≈27.1%) Decreased methanol permeability (50%) Highest proton conductivity: 9.40 × 10^−2^ S cm^−1^ at 110 °C	DMFC	[28]
CS/functionalized silica sub-microspheres	Functionalized silica sub-microspheres	Functionalized silica sub-microsphere synthesis using distillation–precipitation polymerization Solution-casting method membrane	Increased water uptake (45.3–54.5%) Reduced methanol permeability (3.77 × 10^−7^ cm^2^ s^−1^) at 12 M methanol feed Highest ion exchange capacity: 0.493 mmol g^−1^ Highest proton conductivity: 0.029 S cm^−1^	DMFC	[42]
CS-organophosphorylated titania sub-microspheres (CS-OPTi)	Organophosphorylated titania sub-microspheres (CS-OPTi)	OPTi synthesis using modified sol–gel method Solution-casting method membrane	Decreased water uptake (≈10%) Reduced swelling properties (≈10%) Increased ion-exchange capacity (0.096 –0.206 mmol g^−1^) Highest proton conductivity: 11.4 mS cm^−1^ Decreased methanol permeability (2.80 × 10^−7^ cm^2^ s^−1^) at 12 M methanol feed	DMFC	[43]
Sorbitol-plasticized CS/zeolite hybrid	Zeolite (mordenite)	Incorporation of sorbitol into CS/mordenite matrix Solution-casting method membrane	Increased water uptake Lowest methanol permeability: 4.90 × 10^−7^ cm^2^ s^−1^ in 12 M methanol at 25 °C	DMFC	[35]
Surface-modified Y zeolite-filled CS	Modified Y zeolite	Incorporation of APTES and MPTMS on Y zeolite Solution-casting method membrane	Lowered water uptake Reduced ion exchange capacity Lowest methanol permeability: 3.90 × 10^−7^ cm^2^ s^−1^ in 12 M methanol Highest proton conductivity: 2.58 × 10^−2^ S cm^−1^	DMFC	[34]
CS/zeolite hybrid	Zeolite	Incorporation of zeolite into the CS matrix Solution-casting method membrane	Enhanced mechanical strength Increased water uptake (hydrophilic zeolite) Decreased water uptake (hydrophobic zeolite) Reduced methanol permeability Decreased proton conductivity	DMFC	[36]
CS/zeolite beta hybrid	Zeolite beta	Zeolite beta synthesis using hydrothermal method Solution-casting method membrane	Decreased water uptake Reduced ion exchange capacity Lowest methanol permeability: 2.46 × 10^−6^ cm^2^ s^−1^ at 12 M methanol Reduced proton conductivity	DMFC	[17]
CS/zeolite beta hybrid	Zeolite beta	Zeolite beta synthesis using hydrothermal method, functionalized by GPTMS Solution-casting method membrane	Enhanced mechanical strength (104.5 MPa) Decreased water uptake Lowest methanol permeability: 2.20 × 10^−7^ cm^2^ s^−1^ at 12 M methanol Reduced proton conductivity	DMFC	[37]

**Table 2 ijms-21-00632-t002:** HPA-related CS hybrid membranes.

**Membrane**	**Heteropoly Acids**	**Proton Conductivity**	**Performance**	**Application**	**References**
PEC membrane (CS and phosphotungstic acid)	Phosphotungstic acid, H_3_PW_12_O_40_	0.024 S cm^−1^ at 80 °C	Higher swelling properties than Nafion membrane Low methanol permeability: 3.30 × 10^−7^ cm^2^ s^−1^	DMFC	[65]
CS/HPA composite membrane	Phosphomolybdic acid, H_3_PMo_12_O_40_; phosphotungstic acid, H_3_PW_12_O_40_; silicotungstic acid, H_4_SiW_12_O_40_	0.15 cm^−1^ at 25 °C	Increased water uptake (18.6%) Low methanol permeability: 2.70 × 10^−7^ cm^2^ s^−1^ Highest selectivity: 5.60 × 10^4^ S cm^−3^ s	DMFC	[51]
PEC membrane (CS and phosphotungstic acid with montmorillonite)	Phosphotungstic acid, H_3_PW_12_O_40_ and MMT	0.030 S cm^−1^ at 25 °C	Decreased water uptake Reduced methanol permeability Highest power density: 49.7 mW cm^−2^ at 70 °C	DMFC	[54]
Cesium phosphotungstate salt and CS membrane (CTS/Cs_2_-PTA)	Cesium phosphotungstate salt, Cs_2_-PTA	6.00 × 10^−3^ S cm^−1^ at 25 °C and 1.75 × 10^−2^ S cm^−1^ at 80 °C	Increased water uptake Enhanced mechanical strength (≈50 MPa) Lowest methanol permeability: 5.60 × 10^−7^ cm^2^ s^−1^ Highest selectivity: 1.10 × 10^4^ S cm^−3^ s	DMFC	[66]
CS-phosphotungstic acid complex membrane	Phosphotungstic acid, H_3_PW_12_O_40_	≈18 mS cm^−1^	Open circuit potential: 0.95 V at 25 °C Highest power density: 350 mW cm^−2^ at 25 °C	PEMFC	[67]
Anodisc-supported CS/phosphotungstic acid membrane	Phosphotungstic acid, H_3_PW_12_O_40_	≈14 mS cm^−1^	Highest power density: 550 mW cm^−2^	PEMFC	[57]
Sub-micropore CS/phosphotungstic acid membrane (*smp*CTS/HPW)	Phosphotungstic acid, H_3_PW_12_O_40_	2.90 × 10^−2^ S cm^−1^ at 80 °C	Increased water uptake Improved mechanical strength Reduced methanol permeability: 4.70 × 10^−7^ cm^2^ s^−1^ Highest selectivity: 2.27 × 10^4^ S cm^−3^ s Max power density: 16 mW cm^−2^ at 80 °C	DMFC	[68]
Mixed phosphotungstic/phosphomolybdic acid CS membrane (CS/PMA-PTA)	Phosphomolybdic acid, H_3_PMo_12_O_40_ and phosphotungstic acid, H_3_PW_12_O_40_	≈7 mS cm^−1^	Max power density: 350 mW cm^−2^	PEMFC	[58]
Cesium-substituted mesoporous phosphotungstic acid/CS membrane (CS/m-PTA)	Cesium-substituted mesoporous phosphotungstic acid (m-PTA)	1.85 × 10^−2^ S cm^−1^	Increased water uptake Improved mechanical strength Reduced methanol permeability (35.4%) Max power density: 83 mW cm^−2^ with 2 M methanol feed	DMFC	[59]
Sulphonated CS-PEO/HPA membrane	Phosphomolybdic acid, H_3_PMo_12_O_40_; phosphotungstic acid, H_3_PW_12_O_40_	9.21 × 10^−2^ S cm^−1^ at 80 °C	Increased water uptake Max power density: 88.7 mW cm^−2^ at 30 °C	PEMFC	[69]
Sulphonated CS-PEO-s-SiO_2_ doped phosphotungstic acid membrane	Phosphotungstic acid, H_3_PW_12_O_40_	1.53 × 10^−1^ S cm^−1^ at 25 °C	Increased water uptake (48.72–71.42%) Improved tensile strength (16.01–22.78 MPa)	PEMFC	[70]

**Table 3 ijms-21-00632-t003:** Applications of carbon nanotubes (CNTs) in CS hybrid membranes.

**Membrane**	**Fillers**	**Proton Conductivity**	**Performance**	**Application**	**References**
CS-functionalized MWCNTs	MWCNTs	n/a	Increased water absorption Open circuit voltage: 0.75 V Max current density: 150 mA m^−2^ Max power density: 46.94 mW m^−2^	MFC	[94]
CS/silica-coated CNTs	Silica-coated CNTs	0.015–0.025 S cm^−1^	Improved tensile strength (17.8–32.9 MPa) Decreased water uptake (136–100%)	PEMFC	[86]
CS/titania-coated CNTs	Titania-coated CNTs (TCNTs)	0.016–0.023 S cm^−1^	Decreased water uptake (124–74%) Enhanced tensile strength (17.8–29.0 MPa)	PEMFC	[95]
CS/CS-coated CNTs	CS-coated CNTs	9.70 × 10^−3^ S cm^−1^ at 20 °C 3.46 × 10^−2^ S cm^−1^ at 80 °C	Improved tensile strength (29.34–47.91 MPa) Increased water uptake (93.88–108.28%) Open circuit voltage: 0.72 V Max power density: 47.5 mW cm^−2^ at 5 M methanol concentration at 70 °C	DMFC	[83]
CS/CNT fluids	Solvent-free CNT fluids	0.044 S cm^−1^ at 80 °C	Improved tensile strength (35.7–63.9 MPa) Decreased water uptake (78.8–73.6%) Open circuit voltage: ≈0.7 V Max power density: 48.46 mW cm^−2^ at 2 M methanol concentration	DMFC	[96]
CS/superacidic sulphated zirconia-coated CNTs	Superacidic sulphated zirconia-coated CNTs	3.40 × 10^−2^ S cm^−1^ at 80 °C	Enhanced tensile strength (50% greater) Slight increase in water uptake Max power density: 64.6 mW cm^−2^ at 70 °C	DMFC	[93]
CS/ionized organic compounds/hydroxylated MWCNTs	*N*-Benzyl-*N*,*N*-dimethyl-3-((2-methyl-1,3-dioxo-2,3-dihydro-1H-benzo[de] isoquinolin-6-yl) amino) propan-1-aminium hydroxide and hydroxylated MWCNTs-OH	0.83–5.66 × 10^−3^ S cm^−1^	Improved tensile strength (24.17–33.48 MPa) Increased water uptake (0.74–1.12 g g^−1^) Increased ion exchange capacity (0.13–0.69 mequiv g^−1^) Open circuit voltage: ≈0.96 V Max current density: 59.9 mA cm^−2^ Max power density: 31.6 mW cm^−2^	PEMFC	[97]
Sodium lignin sulphonate (SLS) doped TCNTs CS membrane	Anatase TCNTs and sodium lignin sulphonate	3.67 × 10^−2^ S cm^−1^ at 25 °C 6.47 × 10^−2^ S cm^−1^ at 60 °C	Enhanced tensile strength (16.05–23.12 MPa) Increased water uptake (132.95–157.44%) Increased ion exchange capacity (0.16–0.38 mmol g^−1^) Highest selectivity: 28.2 × 10^4^ S cm^−3^ s	DMFC	[98]
CS/sulphonated MWCNTs	PS@CNT (sulphonated by 1, 3-propane sultone; PS method) and DP@CNT (distillation-precipitation polymerization; DP method)	0.011–0.026 S cm^−1^	Improved tensile strength (37.1–51.0 MPa) Decreased water uptake (81–61%) Increased ion exchange capacity (0.18–0.33 mmol g^−1^)	PEMFC	[99]
CS/polydopamine-functionalized CNTs	Polydopamine- functionalized CNTs	0.028 S cm^−1^ at 80 °C	Enhanced tensile strength (20.8–30.5 MPa) Reduced water uptake	DMFC	[100]

n/a means that there is no data available from the articles.

**Table 4 ijms-21-00632-t004:** Applications of graphene oxide (GO) in CS hybrid membranes.

**Membrane**	**Fillers**	**Proton Conductivity**	**Performance**	**Application**	**References**
CS/GO nanocomposites	GO	n/a	Improved mechanical properties (40.1–89.2 MPa)	n/a	[109]
GO cross-linked CS (CS nanocomposite)	GO	n/a	Enhanced tensile strength (43.2–104.2 MPa)	n/a	[110]
CS/SGO	SGO	0.0612 S cm^−1^ at 85 °C (100% RH) 10.9 mS cm^−1^ at 120 °C (0% RH)	Improved tensile strength (44.7–85.3 MPa) Reduced water uptake (76.7–59.7%) Increased ion exchange capacity (0.195–0.223 mmol g^−1^) Open circuit voltage: 0.99 V Max current density: 459.3 mA cm^−2^ Max power density: 146.7 mW cm^−2^	PEMFC	[52]
*N*-*o*-sulphonic acid benzyl CS/*N*,*N*-di-methylene phosphonic acid propylsilane (NSBC/NMPSGO)	NMPSGO	8.87 × 10^−2^ S cm^−1^ at 30 °C (100% RH)	Improved water uptake (69.98%) Increased ion exchange capacity (2.09 mequiv g^−1^) Reduced methanol permeability (16.93 × 10^−7^ cm^2^ s^−1^)	DMFC	[62]
Montmorillonite/GO/CS composite	GO	n/a	Enhanced tensile strength (27.00 MPa)	n/a	[117]
CS/phosphorylated GO	PGO	63.4 mS cm^−1^ at 95 °C (100% RH) 5.79 mS cm^−1^ at 160 °C (0% RH)	Improved tensile strength (44.7–51.5 MPa) Decreased water uptake (53.7–42.1%) Increased ion exchange capacity (0.79 mmol g^−1^) Open circuit voltage: 0.99 V Max current density: 332.5 mA cm^−2^ Max power density: 107.0 mW cm^−2^	PEMFC	[114]
Modified-sulphonated CS	MGO	6.77 × 10^−2^ S cm^−1^ at 30 °C 11.20 × 10^−2^ S cm^−1^ at 90 °C	Improved mechanical strength (21.64–54.00 MPa) Increased water uptake (27–43%) Enhanced ion exchange capacity (1.56–2.56 mequiv g^−1^) Reduced methanol permeability (1.01 × 10^−6^ cm^2^ s^−1^) Highest selectivity: 1.26 × 10^4^ S cm^−3^ s	DMFC	[118]
CS/SCS/SGO	SCS/SGO	1.30–7.20 mS cm^−1^	Enhanced tensile strength (72.4–155.8 MPa) Increased water uptake Increased ion exchange capacity (0.65–1.20 mequiv g^−1^ Reduced methanol permeability (4.62 × 10^−8^ cm^2^ s^−1^ Highest selectivity: 15.15 × 10^4^ S cm^−3^ s	DMFC	[111]
Phosphorylated or sulphurized CS- mixed-matrix composite	GO	n/a	Improved tensile strength (25.13 N mm^−2^) Decreased water uptake (97.4–49.3%) Max power density: 181.56 mW m^−3^	MFC	[119]
Sulphonated CS/polyethylene oxide/sulphonated GO	SGO	4.83 × 10^−2^ S cm^−1^ at 30 °C 11.11 × 10^−2^ S cm^−1^ at 80 °C	Increased water uptake (38.27–56.30%) Increased ion exchange capacity (0.34–0.67 mequiv g^−1^)	PEMFC	[112]

n/a means that there is no data available from the articles.

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
