# Peer review of "(untitled)"

_ijms, 2020, doi:10.3390/ijms21020632_

Round 1
Reviewer 1 Report
First, the title needs to be changed. It is not clear if the content regards chitosan- based polymers and also ionic liquids or chitosan-based polymers and chitosan-based ionic liquids. Most probably, it is the latter, but then the title would be incorrect because there are no chitosan-based ionic liquids! We can maybe talk about chitosan-supported ionic liquids.
In general, this review is overloaded with information copied from numerous papers. For some reader the wealth of data might be valuable. Unfortunately, other readers may have difficulties in judging what is the best strategy for chitosan modification that will lead to a reasonably performing PEM. In fact, based on my knowledge, there are NO demonstrations of chitosan-based PEM that would come even close to Nafion or even to a moderately sulfonated PEEK, regarding combined ionic conductivity, mechanical strength and oxidative stability. The authors should not only copy the content of numerous papers but also attempt to draw some conclusions as to what is the most promising modification path from CS to CS-based-PEM that could power future fuel cells.
These are my more specific comments:
Line 30 – “The current energy crisis due to the deficiency and depletion of natural resources…” – There is NO energy crisis right now. At least, NO global energy crisis that I know about.
Line 32 – “These cells are new materials…” – Fuel cells are NOT materials; they are made from materials. This part of the sentence should be deleted.
Line 34 – “They generate electricity by converting chemical energy from fuel through a chemical reaction.” – Not really! It is an electrochemical reaction or rather coupled electrochemical reactions.
Line 38 – “…thin permeable sheet…” – It should be: thin, ion-permeable sheet or thin, proton-permeable sheet.
Line 40 – “high power density, which accelerates start-ups.” – This sentence is generally not true and should be deleted. If anything, self-heating due to polarization losses helps in reaching the target (elevated) cell temperature.
Figure 1 requires modification. The catalyst layers are not in the form of separate blobs. They form thin films between the GDLs and the membrane and are thinner than the GDLs.
Line 51 – “…oxygen diffusion from the cathode to the anode.” – Crossover of hydrogen is even more important problem.
Line 51 – “Nafion also limits the operation of PEMFC to low temperatures (~80 °C), because it needs hydration and permanent gas humidification to assure high proton conductivity. This requirement has prompted researchers to study new environmentally friendly and economical PEM materials.” – The problem of membrane dehydration is being addressed by developing membranes that do not require hydration or are less sensitive to hydration, not necessarily from “environmentally friendly and economical” materials.
Line 68 – “CS is a natural polymer that is inert…” - Chitosan is a diverse class of polymers MANUFACTURED from a variety of raw materials. Chitosan has reactive -OH and -NH2/-NH3+ groups so it cannot be considered inert.
Line 80 – “…and electrical conductivity;” – Good PEM should have negligible “electrical” or rather electronic conductivity. The authors are probably thinking about ionic (proton/protonic) conductivity.
Line 89 – “Nonmodified CS membranes have high hydrophilic properties that cause a high degree of swelling and reduce sturdiness in fuel cell applications by increasing water uptake and fragility” but then, in Line 114, “The addition of hygroscopic oxides, e.g. Al2O3, TiO2, ZrO2 and (SiO2)n, into the polymer matrix enhances the water retention capacity and ionic conductivity of 116 polymer composites at low RH [12-14]. So, is inherent hydrophilicity of CS not enough? I think the authors should restructure the review in such a way that they first present a problem, e.g., low conductivity, and then present literature data on ways to overcome the limitation, here on how to increase the ionic conductivity of CS. The other key characteristics would be: gas crossover, mechanical strength, retention of conductivity at low RH and elevated temperature, and hydrolytic/oxidative stability.
Table 2 – “Better swelling properties than Nafion membrane” – What does this means?
Table 2 – The bottom entry: “Free-standing CS/phosphotungstic acid membrane.” No, the membrane was really an Anodisc-supported CS/PTA composite, so it was rather Anodisc/CS/PTA membrane. Also, the fuel cell tests were done using pure oxygen instead of the usual oxidant – air, and the catalyst loading was very high, that’s the reason for somewhat high FC power output with this relatively low-conductivity membrane (14 mS/cm).
Line 434 – “…was greater than that of the pristine CS membrane (2.5 × 10-2 S cm-1).” – It is impossible for the pristine, clean CS to have this high conductivity. I think that conductivity should be on the order of 10^(-4) S/cm but not 10^(-2) S/cm. The authors should carefully select references for their review.
Line 527 – “A new improved CS/polymer composite membrane can be formed when synthetic and natural polymer membranes are combined with CS membranes. Two major approaches are used to prepare CS/polymer blends,” – Composite is a different polymer system than blend. The authors should be more careful when naming polymer systems.
In summary, this is an interesting review of the existing literature on fuel cell membranes utilizing CS but it lacks criticism. The truth is that CS is not inherently conductive enough so it needs to be chemically modified. Yes, it is an interesting material for numerous modifications but taking into account highly oxidative environment of the fuel cell catalytic environment, the not-that-robust CS chemical structure may never prove to be a good fit for PEMs (I mean long term performance and not 1 hour laboratory test). The authors did not show even a single really successful (and FC stable) realization of CS-based PEM, or am I wrong?
Author Response
First, the title needs to be changed. It is not clear if the content regards chitosan- based polymers and also ionic liquids or chitosan-based polymers and chitosan-based ionic liquids. Most probably, it is the latter, but then the title would be incorrect because there are no chitosan-based ionic liquids! We can maybe talk about chitosan-supported ionic liquids.
Response:
Thank you for the valuable input.
We have changed the title of the review paper as suggested.
“Title: Review of Chitosan-Based Polymers as Proton Exchange Membranes and Roles of Chitosan-Supported Ionic Liquids”
In general, this review is overloaded with information copied from numerous papers. For some reader the wealth of data might be valuable. Unfortunately, other readers may have difficulties in judging what is the best strategy for chitosan modification that will lead to a reasonably performing PEM. In fact, based on my knowledge, there are NO demonstrations of chitosan-based PEM that would come even close to Nafion or even to a moderately sulfonated PEEK, regarding combined ionic conductivity, mechanical strength and oxidative stability. The authors should not only copy the content of numerous papers but also attempt to draw some conclusions as to what is the most promising modification path from CS to CS-based-PEM that could power future fuel cells.
Response:
We have added some statements in the review paper:
“Previous works by various researchers have proven that the CS composite membrane with unmodified inorganic hygroscopic oxide fillers possess low proton conductivity, water retention capacity and mechanical stability of the membrane. Thus, these drawbacks can be solved through the presence of modified hygroscopic oxide fillers, such as functionalised silica and zeolite, in CS hybrid membranes, due to hydrogen bond existed between the CS polymer and the fillers, as well as the presence of the zwitterionic groups in the fillers, which provide additional proton-conducting pathway in the membrane. These occurrences have suppressed fuel crossover and improved the morphology, proton conductivity, water retention capacity, and mechanical strength and consequently enhanced fuel cell performance of the composite membranes..”
“The high solubility of HPAs in electrochemically produced water promotes leakage and decreases performance, but mixing the HPAs into a CS matrix can alleviate these issues. Besides that, the fuel cell performance and proton conductivity, as well as the mechanical properties of the CS-HPAs composite membranes can be further improved as the HPAs are functionalised by adding inorganic fillers and mesoporous nature onto them, which have the roles of providing proton conduction sites and the degree of dispersion between the polymer matrix and the modified HPAs that led to the formation of homogeneous hybrid polymer membranes..”
“The poor dispersion of CNTs in the polymer matrix was caused by strong Van der Waals forces, which can be modified by altering the surfaces of the CNTs. CNTs that are functionalised with inorganic materials have proven their ability in providing additional proton-conducting pathway, which helped in enhancing the proton conductivity of the membrane. Other than that, this modification considerably improved the compatibility of CNT fillers with the polymer matrix, producing homogeneous dispersion of CNT-polymer mixture, thus enhanced the mechanical stability and fuel cell performance of the composite membranes..”
“In conclusion, previous studies showed that the usage of GO as a filler in CS polymer matrix formed well-dispersed CS-GO composite membranes with good mechanical properties. However, the mechanical properties of the composite membrane can be further improved due to the homogeneous dispersion mixture by adding functionalised GO into the CS polymer matrix. Other than that, this modification included the introduction of acid groups, which acted as proton carriers, providing hopping sites for proton migration in the membranes, enhancing their proton conductivity and fuel cell performance..”
“It can be concluded that as CS polymers are blended with other synthetic polymers, particular properties of the produced blended membranes, such as mechanical stability, proton conductivity and water uptake can be improved. However, there are certain polymers that possessed hydrophobic properties, which caused reduction in proton conductivity and ion exchange capacity of the blended membranes. These drawbacks can be promoted by functionalising the polymers with sulfonic acid groups and as the blended membranes undergo through cross-linking process, the mechanical stability, proton conductivity, water retention capacity and ion exchange capacity of the blended membranes are further improved.”
“As conclusion, various methods of modifications on CS-based membranes have been performed by researchers, including sulfonation, phosphorylation, quaternisation and cross-linking methods. Each of the methods have shown their ability to improve the efficiency and fuel cell performance of the membranes. CS composite membranes that have been modified using sulfonation method, in which the CS polymer is functionalised with sulfonic groups exhibited enhancement in proton conductivity, due to the proton hopping sites available for proton transfer to occur, without improving their solubility. Meanwhile, as the CS membranes are modified through phosphorylation and quaternisation method, it has been reported by previous research that both phosphonic groups and quaternary ammonium groups in each of the methods, respectively, have the ability to enhance the solubility of CS membranes, reduce their crystallinity, increase their hydrophilicity properties, and hence enhance the proton conductivity of the membranes as there are presence of zwitterionic activity and proton transfer in the membranes. Other than that, the CS composite membranes can be further improved when they are cross-linked with cross-linking agents such as sulphuric acid, GA, SSA, etc. However, according to previous works, as the CS composite membranes were cross-linked with binary cross-linking agents (GA and SSA), the achieved proton conductivity was much higher than the individual cross-linking agent, due to the supplementary proton-conducting pathway developed for proton migrations in the membrane. All of these methods have also proven to improve the mechanical strength of the membranes and based on previous reported works, it can be observed that the most promising method of improving the membranes is by functionalising the CS polymer and consequently cross-linking the CS composite membranes, to be used as PEM in fuel cell applications..”
Line 30 – “The current energy crisis due to the deficiency and depletion of natural resources…” – There is NO energy crisis right now. At least, NO global energy crisis that I know about.
Response:
Thank you for the valuable input. We agree with the reviewer’s comment and change the sentence to:
The occurrence of global warming is due to the carbon dioxide emissions, which leads to earth’s climate change and consequently affects the environment, health and economy.”
Line 32 – “These cells are new materials…” – Fuel cells are NOT materials; they are made from materials. This part of the sentence should be deleted.
Response:
Thank you for the valuable input. We agree with the reviewer’s comment and remove the part of sentence “new materials” and keep the sentence of “These cells are devices for power generation that operate in a similar manner as conventional batteries and that require a continuous supply of fuel.”
Line 34 – “They generate electricity by converting chemical energy from fuel through a chemical reaction.” – Not really! It is an electrochemical reaction or rather coupled electrochemical reactions.
Response:
Thank you for the valuable input. We agree with the reviewer’s comment and change the sentence to:
“They generate electricity by using fuel source such as hydrogen through electrochemical reaction.”
Line 38 – “…thin permeable sheet…” – It should be: thin, ion-permeable sheet or thin, proton-permeable sheet.
Response:
Thank you for the valuable input. We agree with the reviewer’s comment and add the word “ion” into the sentence of
“…of thin, ion-permeable sheet…”
Line 40 – “high power density, which accelerates start-ups.” – This sentence is generally not true and should be deleted. If anything, self-heating due to polarization losses helps in reaching the target (elevated) cell temperature.
Response:
Thank you for the valuable input. We agree with the reviewer’s comment and change the sentence to:
“They operate at relatively low temperatures (~80 °C), have high power density and suitable for various applications.”
Figure 1 requires modification. The catalyst layers are not in the form of separate blobs. They form thin films between the GDLs and the membrane and are thinner than the GDLs.
Response:
Thank you for the valuable input. We agree with the reviewer’s comment and change the sentence to:
“…hydrogen crossover from anode to cathode that reduces fuel efficiency and fuel cell open circuit voltage (OCV).”
Line 51 – “…oxygen diffusion from the cathode to the anode.” – Crossover of hydrogen is even more important problem.
Response:
Thank you for the valuable input. We agree with the reviewer’s comment and change the sentence to:
“…hydrogen crossover from anode to cathode that reduces fuel efficiency and fuel cell open circuit voltage (OCV).”
Line 51 – “Nafion also limits the operation of PEMFC to low temperatures (~80 °C), because it needs hydration and permanent gas humidification to assure high proton conductivity. This requirement has prompted researchers to study new environmentally friendly and economical PEM materials.” – The problem of membrane dehydration is being addressed by developing membranes that do not require hydration or are less sensitive to hydration, not necessarily from “environmentally friendly and economical” materials.
Response:
Thank you for the valuable input. We agree with the reviewer’s comment and change the sentence to:
“…because it needs hydration and permanent gas humidification to assure high proton conductivity and as the temperature is increased above 100 °C, the water affinity and mechanical strength of the membrane will also be reduced. These limitations of Nafion membrane have prompted researchers to study and produce new environmentally friendly and economical PEM materials from renewable sources, with improved chemical and thermal stability, water uptake and mechanical properties, that can potentially replace the synthetic polymers in fuel cell applications.”
Line 68 – “CS is a natural polymer that is inert…” - Chitosan is a diverse class of polymers MANUFACTURED from a variety of raw materials. Chitosan has reactive -OH and -NH2/-NH3+ groups so it cannot be considered inert.
Response:
Thank you for the valuable input. We agree with the reviewer’s comment and remove the word “inert” in the sentence of
“CS is a natural polymer that is hydrophilic, biodegradable and biocompatible.”
Line 80 – “…and electrical conductivity;” – Good PEM should have negligible “electrical” or rather electronic conductivity. The authors are probably thinking about ionic (proton/protonic) conductivity.
Response:
Thank you for the valuable input. We agree with the reviewer’s comment and change the sentence to “…and proton conductivity; …”
Line 89 – “Nonmodified CS membranes have high hydrophilic properties that cause a high degree of swelling and reduce sturdiness in fuel cell applications by increasing water uptake and fragility” but then, in Line 114, “The addition of hygroscopic oxides, e.g. Al2O3, TiO2, ZrO2 and (SiO2)n, into the polymer matrix enhances the water retention capacity and ionic conductivity of 116 polymer composites at low RH [12-14]. So, is inherent hydrophilicity of CS not enough? I think the authors should restructure the review in such a way that they first present a problem, e.g., low conductivity, and then present literature data on ways to overcome the limitation, here on how to increase the ionic conductivity of CS. The other key characteristics would be: gas crossover, mechanical strength, retention of conductivity at low RH and elevated temperature, and hydrolytic/oxidative stability.
Response:
Thank you for the valuable input. We agree with the reviewer’s comment and we have restructured the sentences of:
“Nonmodified CS membranes have high hydrophilic properties that cause a high degree of swelling and reduce sturdiness in fuel cell applications by increasing the water uptake excessively and affecting their fragility [10]. Other than that, in contrast to Nafion membranes, which have high mechanical and thermal strengths and high proton conductivity, the pristine, unmodified CS membranes possess low proton conductivity due to the absence of mobile hydrogen ions in their structure and the excess swelling properties of the membranes gives negative impact in their mechanical properties. These characteristics limit their function and performance, thus the CS membranes were modified by incorporating the inorganic fillers into them [13].”
“The addition of hygroscopic oxides, e.g. Al2O3, TiO2, ZrO2 and (SiO2)n, into the CS polymer matrix by functionalizing the inorganic fillers enhances the water retention capacity and ionic conductivity of polymer composites at low RH as new proton pathway is formed [18-20]. Other than that, the methanol crossover can be reduced as the swelling behavior of the membrane is suppressed and the mechanical, chemical and thermal properties can also be improved through the inherent interfacial interactions between the inorganic fillers and the CS polymer matrix.”
Table 2 – “Better swelling properties than Nafion membrane” – What does this means?
Response:
Thank you for pointing out the error. We have overlooked the wrong choice of word “better” and we have changed the sentence to “Higher swelling properties than Nafion membrane” [65]
Table 2 – The bottom entry: “Free-standing CS/phosphotungstic acid membrane.” No, the membrane was really an Anodisc-supported CS/PTA composite, so it was rather Anodisc/CS/PTA membrane. Also, the fuel cell tests were done using pure oxygen instead of the usual oxidant – air, and the catalyst loading was very high, that’s the reason for somewhat high FC power output with this relatively low-conductivity membrane (14 mS/cm).
Response:
Thank you for the valuable input. We agree with the reviewer’s comment and change the sentence to “Anodisc-supported CS/phosphotungstic acid membrane”. [57]
Line 434 – “…was greater than that of the pristine CS membrane (2.5 × 10-2 S cm-1).” – It is impossible for the pristine, clean CS to have this high conductivity. I think that conductivity should be on the order of 10^(-4) S/cm but not 10^(-2) S/cm. The authors should carefully select references for their review.
Response:
Thank you for pointing out the error. We agree with the reviewer’s comment and change the value of conductivity in the sentence of:
“The highest proton conductivity at 80 °C of the CS/SZr@CNT membrane with 0.5 wt.% SZr@CNT content was 3.40 × 10-2 S cm-1, which was greater than that of the pristine, unmodified CS membrane (~10-4 S cm-1).”
Line 527 – “A new improved CS/polymer composite membrane can be formed when synthetic and natural polymer membranes are combined with CS membranes. Two major approaches are used to prepare CS/polymer blends,” – Composite is a different polymer system than blend. The authors should be more careful when naming polymer systems.
Response:
Thank you for pointing out the error. We agree with the reviewer’s comment and remove the word “composite” in the sentence of “A new improved CS/polymer blend membrane can be formed when synthetic and natural polymer membranes are combined with CS membranes.”
In summary, this is an interesting review of the existing literature on fuel cell membranes utilizing CS but it lacks criticism. The truth is that CS is not inherently conductive enough so it needs to be chemically modified. Yes, it is an interesting material for numerous modifications but taking into account highly oxidative environment of the fuel cell catalytic environment, the not-that-robust CS chemical structure may never prove to be a good fit for PEMs (I mean long term performance and not 1 hour laboratory test). The authors did not show even a single really successful (and FC stable) realization of CS-based PEM, or am I wrong?
Response:
Thank you for the valuable input.
We have added some statements regarding the oxidative stability of the membranes. After further reading on literature reviews, we have found that most of works regarding CS-based polymer membranes have conducted stability test by using Fenton’s reagent for 1 hour at 80 °C.
“The stability test was performed for 1 hour at 80 °C by using Fenton’s reagent (3% H2O2 aqueous solution containing 3 ppm FeSO4) and has shown that in oxidative condition, the weight loss was inhibited due to the highly cross-linked and compact structure, which led to excellent oxidative stability of the membrane [27].”
“Other than that, the stability test was performed by using Fenton’s reagent (3% H2O2 with 2 ppm FeSO4) for 1 hour at 80 °C and has shown that as the concentration of PANI/SiO2 nanofillers was increased, the weight loss was reduced, and this implied that CS-PANI/SiO2 composite membrane has excellent oxidative stability. Consequently, this membrane was tested in single cell for DMFC application and exhibited the peak density of 56.27 mW
cm-2, higher than the unmodified CS membrane (41.15 mW cm-2).”
“The stability test was performed by immersing the membrane in Fenton’s reagent for 1 hour at 80 °C, which has shown the composite membrane has better oxidative stability than pure unmodified chitosan, which more than 98% weight of the composite membranes were sustained after conducting the test. This occurrence was due to the shielding effect from the polymer chain and cross-linking agent presented in the composite membrane that prevented the free radicals of hydroxyl and hydro peroxyl from attacking the polymer chains.”
“Besides that, stability test was performed by immersing the membranes in Fenton’s reagent for 1 hour at 80°C, which has shown the CS/SCNTs membrane with higher content of SCNTs has better oxidative stability, which was increased by 86% when compared to the pristine CS membrane. This improvement was due to the excellent resistance and restraint effect of the SCNTs on the CS polymer chains.”
“Other than that, the stability test was performed to observe the degradation of the membranes in Fenton’s reagent at 80 °C, which has shown that the CS/CS@CNTs composite membrane with higher content of CS@CNTs has better oxidative stability, enhanced by 62%, indicated that the filler has excellent oxidation resistance and it was compatible with the polymer matrix, as the degradation of the CS polymer chains was inhibited.”
Reviewer 2 Report
The reviewed manuscript entitled: “Roles of Chitosan-Based Polymers & Ionic Liquids in Proton Exchange Membranes: A Review” could be considered as very comprehensive review paper considering contemporary research on chitosan and chitosan-hybrids based materials for proton exchange membranes preparation – both acid and ionic liquids doped. This excellent review will be very useful for the research community working on this specific topic but also to the more wide topic of polymer electrolyte membrane fundament and development.
As a main but only minor drawback of the review could be pointed out the lack of discussion about the polysaccharides based membranes degradation / operational stability and longevity as even the perfluorinated ionomer Nafion type membranes suffer this phenomena, though to a less degree. This could be viewed from both mineral acids dopant acidification degradation mechanism, especially at elevated temperatures and the usual in situ OH radicals generated during fuel cells or electrolysis cell operation. Or at least some notes to be mentioned this particular problem which is essential for all PEM/AEM materials.
114: The addition of hygroscopic oxides, e.g. Al2O3, TiO2, ZrO2 and 115 (SiO2)n, into the polymer matrix enhances the water retention capacity and ionic conductivity of polymer composites at low RH [12-14]” Here I would suggest also reading/citation of Ublekov et al. paper: “Protonated montmorillonite as a highly effective proton-conductivity enhancer in p-PBI membranes for PEM fuel cells” Materials Letters 135, Pages 5-7 (2014), where very high proton conductivities even at low RH are achieved for a PBI membrane.
Figure 1. Schematic of a typical membrane electrode assembly (MEA) and (b) an example of stacked 47 MEA by ELECTROCHEM INC. Manufacture. Unclear – to be corrected.
197: Figure 4. of the proton conductivity of (a) dry and (b) hydrated CS-PAni/SiO2 hybrid membranes against temperature [19]. Please correct the beginning of the sentence and give some explanation about the chosen abreviations PAni instead of the traditional PANI.
Figure 3. conductivity of the pristine CS, CS/HMSS and CS/PHMSS-ATMP hybrid membranes with 156 increasing temperature at 100% RH [21]. Need correction!
Figure 4. of the proton conductivity of (a) dry and (b) hydrated CS-PAni/SiO2 hybrid membranes 198 against temperature [19]. Same
Seems almost all Figures need caption text corrections/improvement!
2.2.2.1.3. CNTs Please write Carbon nanotubes instead of abbreviation and never start a sentence with recently unrevealed abbreviation!
2.2.2.1.4. GO The same remark – Graphene oxide and start the chapter with Graphene oxide (GO)…
2.3.3. Quaternisation – I think this chapter for quaternized derivatives of chitosan is irrelevant to the Review topic which is for proton exchange membranes not for anion-exchange one and better to be removed or explanation given.
Author Response
The reviewed manuscript entitled: “Roles of Chitosan-Based Polymers & Ionic Liquids in Proton Exchange Membranes: A Review” could be considered as very comprehensive review paper considering contemporary research on chitosan and chitosan-hybrids based materials for proton exchange membranes preparation – both acid and ionic liquids doped. This excellent review will be very useful for the research community working on this specific topic but also to the more wide topic of polymer electrolyte membrane fundament and development.
As a main but only minor drawback of the review could be pointed out the lack of discussion about the polysaccharides based membranes degradation / operational stability and longevity as even the perfluorinated ionomer Nafion type membranes suffer this phenomena, though to a less degree. This could be viewed from both mineral acids dopant acidification degradation mechanism, especially at elevated temperatures and the usual in situ OH radicals generated during fuel cells or electrolysis cell operation. Or at least some notes to be mentioned this particular problem which is essential for all PEM/AEM materials.
Response:
Thank you for the valuable input. We agree with the reviewer’s comment and have included the statement of:
“However, biopolymers such as polysaccharides have drawbacks such as their hydrophilicity properties that can lead to excessive water uptake and swelling in the membrane, which can affect its durability and as the biopolymer membranes are incorporated with super acid that is highly soluble in water, they tend to leach out from the membrane, which then will restrict the long-term use in fuel cell operations [5,6]. Other than that, aside from thermal stability, another factor that affected the fuel cell performance is the durability of the membranes in oxidative condition. As the cell test is conducted in strong oxidising environment, the oxidative degradation is a crucial factor in membrane degradation mechanisms due to the decomposition of products of HO* and HOO* radicals initiated from the hydrogen peroxide that has strong oxidising characteristics throughout the fuel cell operation. The oxidative stability of the membranes was evaluated through immersion of membranes in Fenton’s reagent for 1 hour at 80 °C, and membranes that can endure the oxidative environment can be further used during fuel cell operation. Despite of having these drawbacks, polysaccharides have been widely used and studied due to their distinctive properties, abundance, hydrophilicity and can be chemically modified and functionalized to be used as PEM in fuel cell applications.”
114: The addition of hygroscopic oxides, e.g. Al2O3, TiO2, ZrO2 and 115 (SiO2)n, into the polymer matrix enhances the water retention capacity and ionic conductivity of polymer composites at low RH [12-14]” Here I would suggest also reading/citation of Ublekov et al. paper: “Protonated montmorillonite as a highly effective proton-conductivity enhancer in p-PBI membranes for PEM fuel cells” Materials Letters 135, Pages 5-7 (2014), where very high proton conductivities even at low RH are achieved for a PBI membrane.
Response:
Thank you for the valuable input. We agree with the reviewer’s comment and have included the statement of:
“The study regarding polybenzimidazole membranes with functionalized montmorillonite, which is the protonated montmorillonite (MMT-H) has proven that with very high loadings of MMT-H, the proton conductivity of the membranes achieve drastic improvement at 20% RH with the values of 436 mS cm-1 and mechanical strength of 8.4 MPa [21].” based on the suggested reference.
Figure 1. Schematic of a typical membrane electrode assembly (MEA) and (b) an example of stacked 47 MEA by ELECTROCHEM INC. Manufacture. Unclear – to be corrected.
Response:
Thank you for pointing out the error. We agree with the reviewer’s comment and change the figure.
Corrected – Figure 1. (a) Main components of PEMFCs and (b) schematic of a typical membrane electrode assembly (MEA).
197: Figure 4. of the proton conductivity of (a) dry and (b) hydrated CS-PAni/SiO2 hybrid membranes against temperature [19]. Please correct the beginning of the sentence and give some explanation about the chosen abreviations PAni instead of the traditional PANI.
Response:
Thank you for the valuable input. We have corrected the caption for Figure 4.
After further review, we found that the paper that we referred does not follow the standard abbreviation. We changed all “PAni” to “PANI” to follow the standard abbreviation, including in Figure 4.
Figure 3. conductivity of the pristine CS, CS/HMSS and CS/PHMSS-ATMP hybrid membranes with 156 increasing temperature at 100% RH [21]. Need correction!
Response:
Thank you for pointing out the error. We have changed the caption for Figure 3.
Corrected - Figure 3. Proton conductivity of the pristine CS, CS/HMSS and CS/PHMSS-ATMP hybrid membranes with increasing temperature at 100% RH.
Figure 4. of the proton conductivity of (a) dry and (b) hydrated CS-PAni/SiO2 hybrid membranes 198 against temperature [19]. Same
Response:
Thank you for pointing out the error. We have changed the caption for Figure 4.
Corrected - Figure 4. Proton conductivity of (a) dry and (b) hydrated CS-PAni/SiO2 hybrid membranes with increasing temperatures.
Seems almost all Figures need caption text corrections/improvement!
Response:
Thank you for pointing out the errors. We have changed the caption for all figures.
2.2.1.3. CNTs Please write Carbon nanotubes instead of abbreviation and never start a sentence with recently unrevealed abbreviation!
Response:
Thank you for pointing out the error. We have changed the abbreviation to full word.
Corrected - 2.2.1.3. Carbon nanotubes
2.2.1.4. GO The same remark – Graphene oxide and start the chapter with Graphene oxide (GO)…
Response:
Thank you for pointing out the error. We have changed the abbreviation to full word.
Corrected - 2.2.1.4. Graphene oxide
2.3.3. Quaternisation – I think this chapter for quaternised derivatives of chitosan is irrelevant to the Review topic which is for proton exchange membranes not for anion-exchange one and better to be removed or explanation given.
Response:
Thank you for the valuable input. Authors have decided to sustain the subtopic “2.3.3. Quaternisation” as it is one of the main modifications in chitosan biopolymer, despite its application in other fuel cell applications. The subtopic has shown the modifications of CS membranes through quaternisation process, which then are used as PEM, and we have also included reference, which the quaternised membrane is used in H2/O2 condition fuel cell test.
Reviewer 3 Report
The authors reported the Roles of Chitosan-Based Polymers & Ionic Liquids in Proton Exchange Membranes: A Review. Indeed, this review is significant in the field of proton exchange membranes for PEMFCs. This manuscript is well written and described in detail, hence it can be acceptable for publication. Before publication of the manuscript, the following minor concerns need to be addressed.
Authors should elongate the introduction section. Should include additional features chitosan. Figure 6: There is something wrong in caption. Did authors get permission form the journals to reuse the figures. If, then mention that in caption.
Author Response
The authors reported the Roles of Chitosan-Based Polymers & Ionic Liquids in Proton Exchange Membranes: A Review. Indeed, this review is significant in the field of proton exchange membranes for PEMFCs. This manuscript is well written and described in detail, hence it can be acceptable for publication. Before publication of the manuscript, the following minor concerns need to be addressed.
Authors should elongate the introduction section. Should include additional features chitosan. Figure 6: There is something wrong in caption. Did authors get permission from the journals to reuse the figures. If, then mention that in caption.
Response:
Thank you for the valuable input. We agree with the reviewer’s comment and have included the statement of:
“Polysaccharides, such as chitosan (CS) and cellulose, are among the best natural polymer materials because of their abundance in the environment. CS is a linear polysaccharide consisted of randomly distributed β-(1-4)-linked ᴅ-glucosamine (deacetylated unit) and N-acetyl-ᴅ-glucosamine (acetylated unit), which is produced by the deacetylation of chitin. CS has high water affinity properties as there are presence of three different polar functional groups, which are hydroxyl (-OH), primary amine (-NH2) and C-O-C groups. CS has a structure that is very similar to cellulose, but the difference among them is that CS contains amine groups and its structure is characterized by its molecular weight and degree of deacetylation, where its solubility is depended on [7].”
“CS has rigid structure and high crystallinity as there are three hydrogen atoms strongly bonded between the amino and hydroxyl groups within the CS monomers, due to the immobilized structure, which leads to its proton conduction limitation. Moreover, CS has attractive physicochemical properties in aqueous solution due to its polycationic nature and this leads to wide range of biological purposes such as antimicrobial activity and disease resistance activities in plants [8]. CS membranes are one of the most preferable and favourable materials to replace Nafion membranes as they have shown improved performance in low temperature fuel cell applications and due to their enticing alcohol barrier properties, proton conductivity and thermal stability after they undergo cross-linking process [5, 9].”
Thank you for pointing out the error. We have changed the caption for Figure 6. Corrected - Figure 6. The structure of crosslinked CS-IHPA nanocomposite membranes.
Thank you for pointing that out. Yes, for reviewer’s information, the authors have already received permissions to reuse all figures. We have added copyright and permissions’ statement in the captions for all figures.